# Cryo-EM structures of remodeler-nucleosome intermediates suggest allosteric control through the nucleosome

Jean Paul Armache[1†], Nathan Gamarra[1,2†], Stephanie L Johnson[1], John D Leonard[1,2‡], Shenping Wu[3§], Geeta J Narlikar[1*], Yifan Cheng[1,3*]

[1]Department of Biochemistry and Biophysics, University of California, San Francisco, San Francisco, United States; [2]Tetrad Graduate Program, University of California, San Francisco, San Francisco, United States; [3]Howard Hughes Medical Institute, University of California, San Francisco, San Francisco, United States

**\*For correspondence:**
Geeta.Narlikar@ucsf.edu (GJN);
ycheng@ucsf.edu (YC)

[†]These authors contributed equally to this work

**Present address:** [‡]3T Biosciences, Menlo Park, United States; [§]Yale University, New Haven, United States

**Abstract** The SNF2h remodeler slides nucleosomes most efficiently as a dimer, yet how the two protomers avoid a tug-of-war is unclear. Furthermore, SNF2h couples histone octamer deformation to nucleosome sliding, but the underlying structural basis remains unknown. Here we present cryo-EM structures of SNF2h-nucleosome complexes with ADP-BeF$_x$ that capture two potential reaction intermediates. In one structure, histone residues near the dyad and in the H2A-H2B acidic patch, distal to the active SNF2h protomer, appear disordered. The disordered acidic patch is expected to inhibit the second SNF2h protomer, while disorder near the dyad is expected to promote DNA translocation. The other structure doesn't show octamer deformation, but surprisingly shows a 2 bp translocation. FRET studies indicate that ADP-BeF$_x$ predisposes SNF2h-nucleosome complexes for an elemental translocation step. We propose a model for allosteric control through the nucleosome, where one SNF2h protomer promotes asymmetric octamer deformation to inhibit the second protomer, while stimulating directional DNA translocation.

DOI: https://doi.org/10.7554/eLife.46057.001

## Introduction

ATP-dependent chromatin remodeling motors play central roles in regulating access to the genome (*Clapier and Cairns, 2009*; *Zhou et al., 2016*). Much has been learnt about remodeling mechanisms through the study of four classes of remodeling motors: the SWI/SNF class, the ISWI class, the CHD class and the combined INO80 and SWR class (*Narlikar et al., 2013*). The ATPase subunits of the SWI/SNF, ISWI and CHD classes have been shown to carry out most of the biochemical activities of their parent complexes. Despite sharing sequence homology within their ATPase domains, these motors play distinct roles in vivo and differ significantly in their biochemical activities (*Clapier and Cairns, 2009*; *Narlikar et al., 2013*; *Zhou et al., 2016*). For example, SWI/SNF motors can generate products ranging from translationally repositioned to fully evicted histone octamers (nucleosome sliding and disassembly, respectively). In contrast, the ISWI and CHD family of motors appear to only slide nucleosomes but differ in how their activity is regulated by the extra-nucleosomal DNA flanking a nucleosome and the N-terminal histone H4 tail (*Narlikar et al., 2013*). Finally, while the human ISWI remodeler, SNF2h, functions most optimally as a dimer, SWI/SNF and CHD family remodelers are proposed to mainly function as monomeric ATPases (*Asturias et al., 2004*; *Leonard and Narlikar, 2015*; *Leschziner et al., 2007*; *Qiu et al., 2017*; *Racki et al., 2009*). We note that recent cryo-EM structures of yeast Chd1 showed some states with two Chd1 molecules bound to a nucleosome, but the mechanistic significance of this dimeric architecture is not known (*Sundaramoorthy et al., 2018*).

Despite fundamental mechanistic advances over the past two decades, the structural basis for how remodeling motors work and why different remodeler families differ in mechanism remains poorly understood. Recent advances in electron cryo-microscopy (cryo-EM) methodology have allowed direct visualization of SWI/SNF, CHD, INO80 and SWR remodeling motors bound to the nucleosome at high resolution (*Ayala et al., 2018*; *Eustermann et al., 2018*; *Farnung et al., 2017*; *Liu et al., 2017*; *Sundaramoorthy et al., 2018*; *Sundaramoorthy et al., 2017*; *Willhoft et al., 2018*). Here we present cryo-EM structures of the full-length form of the human ISWI remodeler, SNF2h bound to a nucleosome. Carrying out cryo-EM without any cross-linking and using the ATP analog ADP-BeF$_x$ enabled us to trap three different conformational states of the SNF2h-nucleosome complex: a state with an unexpectedly translocated nucleosome (*Figure 1*, *Figure 1—figure supplements 1–6*), a state with two SNF2h protomers bound to a nucleosome (*Figure 2A*, *Figure 2—figure supplement 1*) and a state with one protomer bound to a nucleosome that shows increased disorder within the histone core (*Figure 2B*). The locations of histone disorder strongly suggest a role for octamer deformation in protomer coordination and directional DNA translocation. In addition, we detect new ISWI-histone contacts that make significant contributions to nucleosome sliding and help explain why ISWI may in differ in mechanism from Swi2/Snf2 (*Figure 3*) (*Liu et al., 2017*).

## Results

### Overview of SNF2h-nucleosome structures

Like most ISWI remodelers, SNF2h slides mono-nucleosomes assembled on short stretches of DNA towards the center of the DNA (*Clapier and Cairns, 2009*; *Narlikar et al., 2013*; *Zhou et al., 2016*). In previous studies we have found that while a monomer of SNF2h can slide nucleosomes, SNF2h functions most optimally as a dimer (*Leonard and Narlikar, 2015*; *Racki et al., 2009*). In these studies, we were able to visualize both singly bound and doubly bound SNF2h using negative stain EM (*Racki et al., 2009*). Previous studies have further shown that binding of the ATP analog, ADP-BeF$_x$, promotes a restricted conformation of the ATPase active site in a manner that is dependent on the H4 tail (*Racki et al., 2014*). The restricted conformation is consistent with observations showing an activating role for the H4 tail (*Clapier et al., 2001*; *Clapier et al., 2002*; *Hamiche et al., 2001*). Further, binding of ADP-BeF$_x$ to SNF2h promotes conformational flexibility of buried histone residues (*Sinha et al., 2017*). This conformational flexibility is functionally important because restricting the flexibility via disulfide bonds inhibits nucleosome sliding (*Sinha et al., 2017*). Based on these observations we have previously reasoned that the ADP-BeF$_x$ state mimics an activated reaction intermediate. With the goal of obtaining high-resolution structures of this intermediate, we assembled SNF2h-nucleosome complexes in the presence of ADP-BeF$_x$. The nucleosomes contain 60 base-pairs (bp) of flanking DNA on one end (0/60 nucleosomes). SNF2h-nucleosome complexes were assembled using conditions similar to those used in our previous negative stain EM experiments with the additional variable of salt concentration as discussed below (*Racki et al., 2009*). Cryo-EM grids were prepared without using cross-linking.

During the course of this study, we collected two cryo-EM datasets using two different salt conditions for optimization of cryo-EM grid preparation. Electron micrographs and two-dimensional (2D) class averages calculated from a cryo-EM dataset collected using lower salt (70 mM KCl) on a scintillator-based camera show a relatively high percentage of doubly bound SNF2h-nucleosome complexes (*Figure 2A*, *Figure 2—figure supplements 1–2A*). In contrast, another dataset collected using higher salt (140 mM KCl) on a K2 direct electron detection camera shows that the majority of the particles have one SNF2h bound to a nucleosome rather than two (*Figure 1*, *Figure 1—figure supplements 1–2*, *Figure 2—figure supplement 2B–C*). The reason for this difference is not fully understood. While higher salt reduces SNF2h affinity for nucleosomes, we believe the increase in salt by itself is not sufficient to cause complex dissociation as by negative stain EM we observe a high proportion of doubly bound complexes under these conditions (*Figure 2—figure supplement 3*). The higher salt concentration may, however, have a bigger impact when combined with other destabilizing factors during the process of plunge freezing cryo-EM grids, some of which are discussed in the Methods and further below.

With the goal of achieving the highest resolution possible we initially focused on the dataset obtained from the K2 direct electron detection camera. Using this dataset we determined a 3D

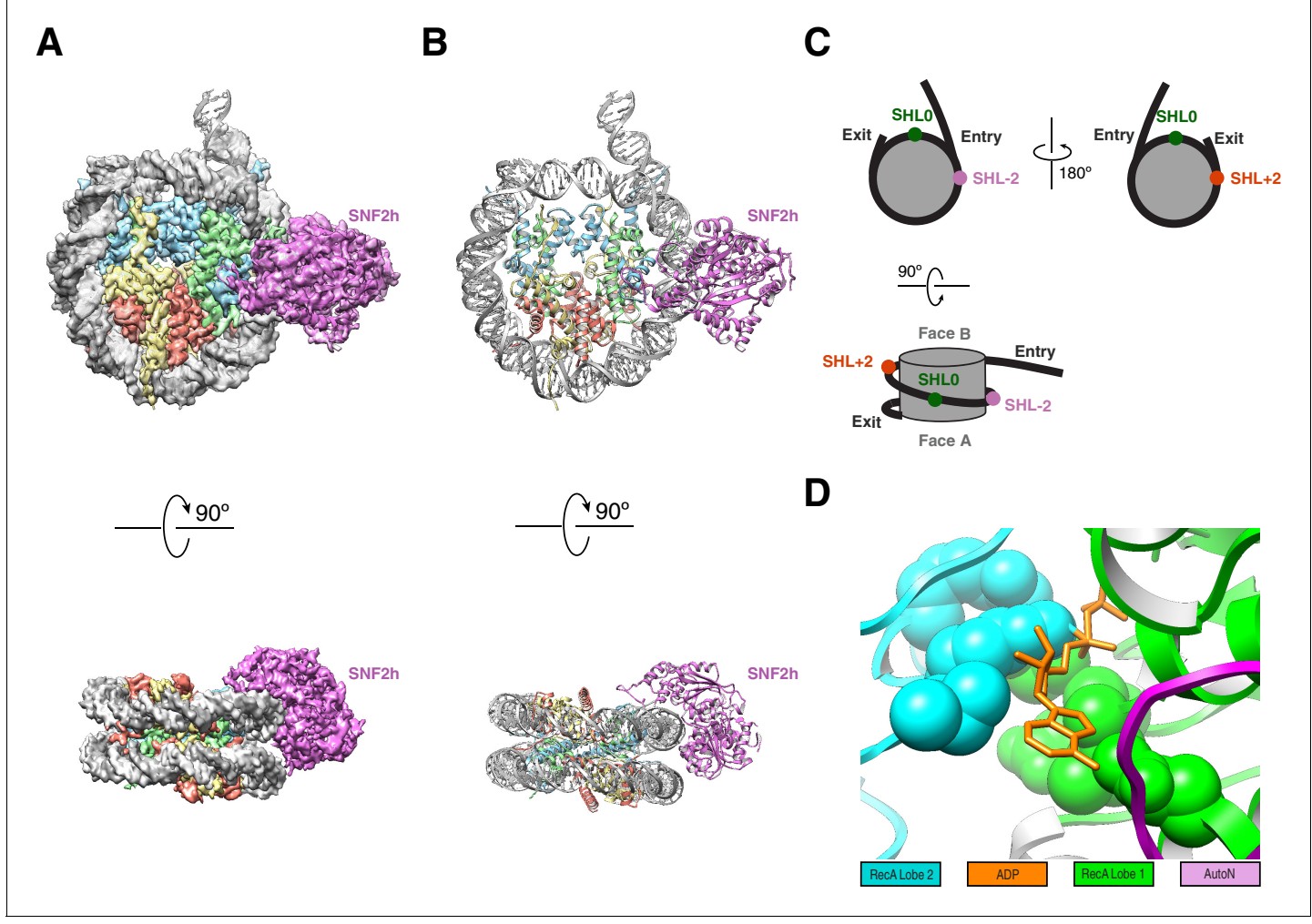

**Figure 1.** High resolution structure of SNF2h bound to a nucleosome with 60 bp of flanking DNA in the presence of ADP-BeF$_x$ and 140 mM KCl. (**A**) Cryo-EM density map of SNF2h bound to the nucleosome at 3.4 Å from data recorded with a K2-summit camera. (**B**) Model built using the density in (**A**). (**C**) Cartoon representation of a nucleosome with asymmetric flanking DNA as in our structures. Super Helical Location (SHL) ± 2 as well as the entry and exit site DNAs are labeled. The SHL0 location is also labeled and is defined as the dyad. Faces A and B of the histone octamer are labeled in gray. (**D**) Zoom into the ATP-binding pocket of SNF2h with ADP in orange and represented with sticks. In spheres are the SNF2h residues that bind nucleotide with the helicase motif I in green and helicase motif VI in blue (***Figure 3—figure supplement 4***).

DOI: https://doi.org/10.7554/eLife.46057.002

The following source data and figure supplements are available for figure 1:

**Figure supplement 1.** Cryo-EM analysis of singly bound SNF2h-nucleosome complexes (140 mM KCl) .

DOI: https://doi.org/10.7554/eLife.46057.003

**Figure supplement 2.** Cryo-EM Densities of SHL+2 and SHL-2 SNF2h-Nucleosome complexes obtained at 140 mM.

DOI: https://doi.org/10.7554/eLife.46057.004

**Figure supplement 3.** Cryo-EM reconstructions of the SNF2h-Nucleosome complexes at 140 mM KCl are translocated ~2 bp.

DOI: https://doi.org/10.7554/eLife.46057.005

**Figure supplement 3—source data 1.** Values used to obtain plots in D.

DOI: https://doi.org/10.7554/eLife.46057.006

**Figure supplement 4.** By a single molecule assay, SNF2h induces a change in FRET under the 140 mM KCl conditions, consistent with a movement of the nucleosomal DNA.

DOI: https://doi.org/10.7554/eLife.46057.007

**Figure supplement 5.** Difference maps to test for extra density of DNA at exit side of SNF2h-nucleosome complexes.

DOI: https://doi.org/10.7554/eLife.46057.008

**Figure supplement 6.** Bootstrapped maps of SNF2h-nucleosome complex.

DOI: https://doi.org/10.7554/eLife.46057.009

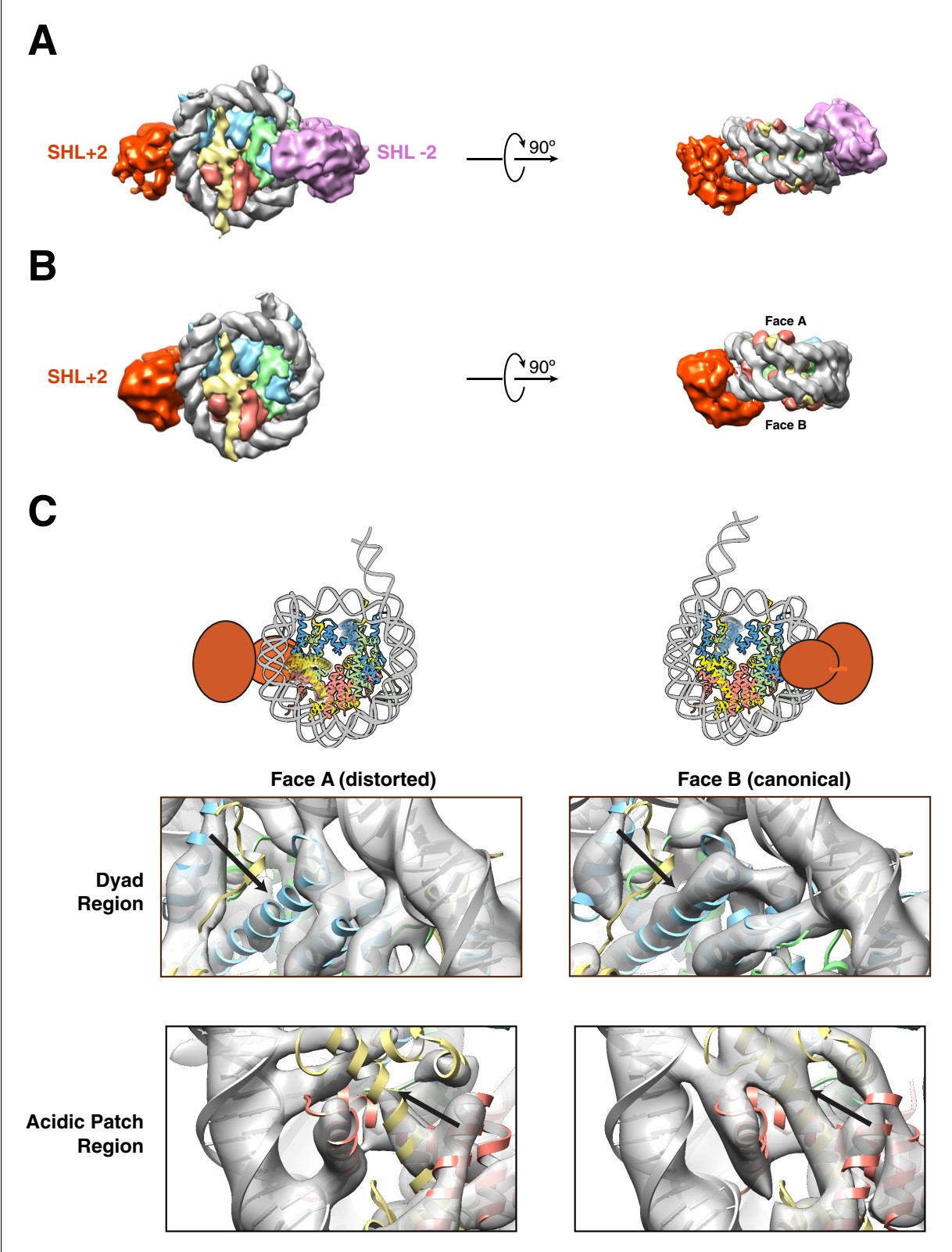

**Figure 2.** Structures of SNF2h bound to a nucleosome with 60 bp of flanking DNA in the presence of ADP-BeF$_x$ and 70 mM KCl. (**A–C**) Cryo-EM density maps of SNF2h bound to the nucleosome recorded with a scintillator-based camera (**A**) Doubly bound SNF2h-nucleosome complex at 8.4 Å resolution. (**B**) Singly bound SNF2h at SHL+2. (**C**) Comparison of the Cryo-EM density on the two faces of the nucleosome. Face A of the nucleosome (left column) has weaker EM density at the histone H2A acidic patch (bottom row) and the α2 helix of H3 (top row) when compared to face B (right column) at the

*Figure 2 continued on next page*

*Figure 2 continued*

same contour level. The black arrows point to the helices that show altered densities in Face A vs. Face B. The regions of increased dynamics are also shown schematically as blurry helices in cartoons of the nucleosome above the densities for Face A and Face B.

DOI: https://doi.org/10.7554/eLife.46057.010

The following figure supplements are available for figure 2:

**Figure supplement 1.** Cryo-EM analysis of doubly bound SNF2h-nucleosome complexes obtained at 70 mM KCl.

DOI: https://doi.org/10.7554/eLife.46057.011

**Figure supplement 2.** 3D Classification and refinement.

DOI: https://doi.org/10.7554/eLife.46057.012

**Figure supplement 3.** Negative stain EM of SNF2h in the presence of ADP-BeF$_x$ and 140 mM KCl.

DOI: https://doi.org/10.7554/eLife.46057.013

reconstruction with a single SNF2h bound to a nucleosome at a resolution of 3.4 Å (*Figure 1*). The majority of particles contributed to this reconstruction having SNF2h bound to the flanking DNA at Super Helical Location (SHL) −2, judging from the density of flanking DNA. The locations of SHL + 2 and −2 as well as the entry and exit site DNA are defined in *Figure 1C*. This map is of sufficient quality for model building of nucleosomal DNA, core histones and the ATPase domain of SNF2h (*Figure 3—figure supplement 1*). The nucleotide binding pocket shows clear density of bound ADP (*Figure 1D*, *Figure 3—figure supplement 2*), but we cannot unambiguously confirm the presence of BeF$_x$. The ATP binding site was also functionally confirmed by mutagenesis (*Figure 3—figure supplement 2*). In addition to the ATPase domain, SNF2h has a C-terminal domain termed HAND-SANT-SLIDE (HSS), which binds flanking DNA, and an N-terminal region termed AutoN, which plays an autoihibitory role (*Figure 3A*) (*Clapier and Cairns, 2012*; *Dang and Bartholomew, 2007*; *Grüne et al., 2003*; *Zhou et al., 2016*). These regions are not visible at high resolution, suggesting conformational flexibility of these regions in this state. By comparison to the K2 dataset, the earlier dataset was collected from a scintillator-based camera, which impeded the achievable resolution of the maps. From this dataset we determined two three-dimensional (3D) reconstructions, one of a nucleosome with doubly bound SNF2h and the other with singly bound SNF2h, both at 8.4 Å resolution with most histone helices fully resolved (*Figure 2A–B*, *Figure 2—figure supplements 1–2A*). The atomic models derived from the 3.4 Å reconstruction fit well into the density for the doubly bound SNF2h-nucleosome complex as rigid bodies (*Figure 2A*).

To assess whether the main difference between the two structures was simply resolution or whether we had trapped different states of the SNF2h-nucleosome complex, we carried out further analysis and comparisons as described below.

## A SNF2h-nucleosome complex that suggests an asymmetrically deformed histone octamer

For the detailed analysis we first focused on the older data set as this contained a larger set of doubly bound particles. Particles contributing to this reconstruction were aligned using flanking DNA as a fiducial marker to break the pseudo symmetry (*Figure 2A*, *Figure 2—figure supplement 2A*). The density of the SNF2h bound at SHL+2 is weaker than that of SNF2h bound at SHL-2 (*Figure 2A*, *Figure 2—figure supplement 2A*). This difference likely suggests that the SNF2h bound to the nucleosome at SHL+2 is conformationally more flexible. Substantial previous work has suggested that when ISWI enzymes move end-positioned nucleosomes towards the center, the active protomer initiates translocation from SHL+2 and engages the entry site flanking DNA via its HSS domain (See *Figure 1C* for nomeclature) (*Dang and Bartholomew, 2007*; *Kagalwala et al., 2004*; *Leonard and Narlikar, 2015*; *Schwanbeck et al., 2004*; *Zofall et al., 2006*). The increased conformational flexibility of the SNF2h protomer bound at SHL+2 is consistent with this protomer being the active one.

Some regions of the histone octamer in the doubly bound structure were less well resolved than other regions, suggesting specific regions of disorder within the octamer (*Figure 2—figure supplement 2G*). The apparent disorder was somewhat symmetric, and without an internal control for comparison, we could not unambiguously interpret the lower resolution as resulting from increased disorder as opposed to achievable resolution. However, we noticed that in the singly bound structures, with SNF2h bound at SHL+2, the disorder is asymmetric providing a chance to use the non-disordered half of the octamer as an internal control for achievable resolution. With this internal

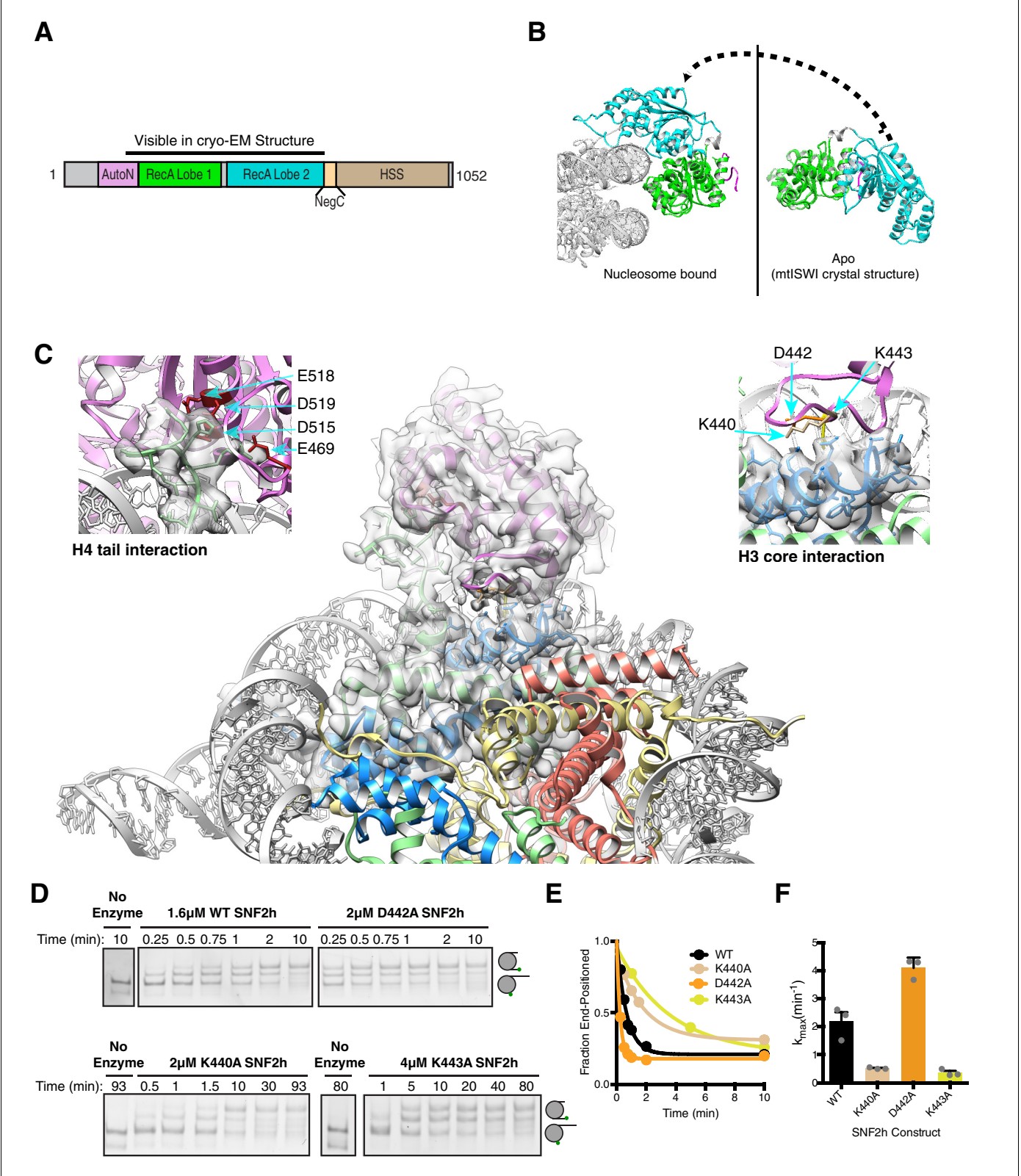

**Figure 3.** Interactions of SNF2h with the histone proteins. (**A**) Domain diagram of SNF2h. (**B**) Conformational changes in SNF2h associated with nucleosome binding. SNF2h is colored according to the domain diagram. The apo structure is the *Myceliophtora thermophila* ISWI crystal structure (PDBID: 5JXR). (**C**) Middle. High resolution SNF2h-nucleosome structure from **Figure 1** enlarged to show details of the interactions with the histone proteins. Colored in red on SNF2h are the acidic residues contacting the histone H4 tail. Colored in orange, tan, and yellow are the residues mutated in

*Figure 3 continued on next page*

*Figure 3 continued*

this study. Left. Enlarged to show details of the H4 tail interaction. Right. Enlarged to show details of the H3 core interaction. (**D**) Native gel remodeling assay of SNF2h constructs. Cy3-DNA labeled nucleosomes were incubated with saturating concentrations of enzyme and ATP and resolved on a native 6% polyacrylamide gel. (**E**) Quantifications of the data in (**D**) zoomed on the x-axis to show effects more clearly and fit to a single exponential decay. Un-zoomed plots are in *Figure 3—figure supplement 6*. (**F**) Rate constants derived from remodeling assays. Bars represent the mean and standard error from three experiments.

DOI: https://doi.org/10.7554/eLife.46057.014

The following source data and figure supplements are available for figure 3:

**Source data 1.** Values plotted in E and F.
DOI: https://doi.org/10.7554/eLife.46057.022
**Figure supplement 1.** Selected Cryo-EM protein densities.
DOI: https://doi.org/10.7554/eLife.46057.015
**Figure supplement 2.** Comparison of ATP-binding pockets of SNF2h with CHD1 and Swi2/Snf2 and functional validation of SNF2h ATP-binding pocket.
DOI: https://doi.org/10.7554/eLife.46057.016
**Figure supplement 3.** Brace helix comparisons.
DOI: https://doi.org/10.7554/eLife.46057.017
**Figure supplement 4.** Multiple sequence alignment of the ATPase domains of selected members of chromatin remodeling families.
DOI: https://doi.org/10.7554/eLife.46057.018
**Figure supplement 5.** ATPase activities of point mutants in this study.
DOI: https://doi.org/10.7554/eLife.46057.019
**Figure supplement 5—source data 1.** Values used to obtain plots.
DOI: https://doi.org/10.7554/eLife.46057.020
**Figure supplement 6.** Full fits of native gel remodeling assays.
DOI: https://doi.org/10.7554/eLife.46057.021

control the reconstruction obtained from the singly bound particles suggests asymmetric deformation of the histone core (*Figure 2B–C*, compare canonical Face B to disordered Face A). These results suggested that (i) the less well resolved local regions in the doubly bound structure also likely result from increased local disorder and (ii) a given SNF2h protomer causes octamer disorder on only one side of the nucleosome. Specifically, two regions of the folded histones show increased disorder (helix α2 in H3 and the H2A/H2B acidic patch, *Figure 2C*). The disorder is apparent when the intact density for the blue, red and yellow helices in face B is compared to the missing or altered density for these helices in face A (*Figure 2C*, black arrows). These locations of octamer disorder are mechanistically informative as detailed below.

The region of increased disorder at helix α2 in H3 is proximal to the nucleosomal dyad. This region also interfaces with buried residues in H4 that showed increased dynamics in our previous NMR studies (*Sinha et al., 2017*). What could be the significance of this potential allosteric deformation? While DNA translocation by SNF2h initiates from SHL+2, nucleosome sliding requires the disruption of histone-DNA contacts to allow propagation of DNA around the octamer. Histone deformation near the dyad could create a relaxed local environment that facilitates disruption and propagation of DNA around the nucleosome. Further, asymmetry in this disruption may facilitate directionality in the sliding reaction. Consistent with this possibility, our previous work shows that constraining the H3 α2 helix by disulfide cross-linking alters the directionality of nucleosome sliding (*Sinha et al., 2017*). Additionally, recent studies by others have suggested asymmetric rearrangements of helix α2 in H3 at 150 mM NaCl and have found that the same disulfide crosslinks inhibit thermally driven nucleosome sliding (*Bilokapic et al., 2018a*). Based on these comparisons, our findings here suggest that SNF2h amplifies intrinsic nucleosome dynamics during the sliding reaction.

The other region of increased disorder is the acidic patch formed between histone H2A and H2B. Previous work has suggested that interactions between SNF2h and the acidic patch play a critical role in stimulating nucleosome sliding (*Dann et al., 2017*; *Gamarra et al., 2018*). Interestingly, recent biochemical studies using asymmetric acidic patch mutant nucleosomes indicate that the activity of a SNF2h protomer bound at SHL+2 (as defined in *Figure 1C*) requires the acidic patch on the undistorted octamer face (Face B) (*Levendosky and Bowman, 2019*). All of these observations raise the intriguing possibility that binding of one SNF2h protomer allosterically deforms the acidic

patch that is required by the second protomer on the other side of the nucleosome. Such an allosteric effect could serve to inhibit the second protomer from initiating sliding in the opposite direction, thus preventing a tug-of-war between the two protomers.

## A SNF2h-nucleosome complex with a translocated nucleosome

The analysis above led us to ask if we could also detect octamer deformation in the newer data set. We first explored if there were particles suggesting increased dynamics in the K2 dataset that we could have missed in the drive for homogeneity and the highest resolution. Including particles with substantial octamer dynamics would by definition increase local disorder in the reconstruction and affect the resolution both locally and globally. However, we failed to extract any subset from the excluded particles that shows signs of octamer dynamics, suggesting that this dataset may not contain particles with deformed octamers.

However, as part of our analysis for detecting octamer dynamics, we re-picked the particles and separated them into two different classes of single SNF2h-bound nucleosomes: a larger one of 3.9 Å resolution with a single enzyme bound at SHL-2 (*Figure 1—figure supplements 1E* and *2B*, *Figure 2—figure supplement 2B*) and a smaller one of 6.9 Å with a single enzyme bound at SHL+2 (*Figure 1—figure supplements 1F* and *2A*, *Figure 2—figure supplement 2B*). The lower resolution of 6.9 Å is primarily due to the small number of particles in this conformation. The atomic models of nucleosomal DNA, core histones and the ATPase domain of SNF2h derived from the 3.4 Å map fit well into the density map of the 3.9 Å and 6.9 Å reconstructions as rigid bodies. Unexpectedly, in both the 3.9 Å and 6.9 Å reconstructions, we observed that 2 bp of DNA is translocated from the exit site (*Figure 1—figure supplements 3* and *5A–B*). The DNA density at the exit side of the nucleosome is intact and fully resolved, suggesting tight association of DNA with the histone octamer, similar to what is observed in other nucleosome structures (*Chua et al., 2016*; *Farnung et al., 2017*). The phosphate groups in the double stranded DNA backbone are clearly resolved, which enabled us to precisely locate and count every bp and to confirm the two extra bp on the exit side of the nucleosome (*Figure 1—figure supplements 3A–B* and *5–6*). We ruled out the possibility that the nucleosome particles were pre-assembled with two bp shifted before forming a complex with SNF2h, by determining a reconstruction of nucleosomes assembled identically but untreated with SNF2h (*Figure 1—figure supplement 5C*, *Figure 2—figure supplement 2D*). These nucleosomes do not display the 2 bp translocation of DNA from the exit side. Further, intriguingly, no extra DNA density was found in the exit side for the structures obtained at 70 mM KCl (*Figure 1—figure supplement 5D*). These comparisons indicate that the reconstructions obtained using the K2 and scintillator-based cameras at 140 mM and 70 mM KCl respectively represent different states of the SNF2h-nucleosome complex.

What could be the significance of the 2 bp translocation? Recent studies have shown that the Chd1 remodeling motor can shift 1–3 bp of nucleosomal DNA inwards from the entry site in the apo and ADP states (*Winger et al., 2018*). These observations raised the possibility that SNF2h may display an analogous property in the presence of ADP-BeF$_x$, the nucleotide analog used in our EM preparations. To test if the ADP-BeF$_x$ bound state causes changes in the conformation of nucleosomal DNA, we used single-molecule FRET (smFRET) experiments to measure changes in the location of exit DNA relative to the histone octamer (*Figure 1—figure supplement 4*). Under the 140 mM KCl buffer conditions of the EM sample preparation, we observed a change in FRET in the presence of SNF2h and ADP-BeF$_x$ (*Figure 1—figure supplement 4D*). The extent of FRET change is consistent with the change in FRET that would be expected from the translocation of 1–2 bp of DNA out of the nucleosome. This FRET change was not observed in the absence of SNF2h (*Figure 1—figure supplement 4D*). Complementary ensemble FRET experiments with a different labeling scheme that reports on changes in distance between the DNA at the exit and entry sites of the nucleosome also showed a change in FRET that is consistent with DNA translocation (*Figure 1—figure supplement 3C–D*). The ensemble FRET change was also dependent on the presence of SNF2h. These results indicate that analogous to Chd1, SNF2h can promote the shifting of ~2 bp of DNA in the absence of ATP hydrolysis. However, these observations raised the question of why comparable DNA translocation was not detected in the EM reconstructions carried out under the 70 mM KCl conditions (*Figure 1—figure supplement 5D*). We reasoned that the combination of SNF2h binding and the higher salt conditions of 140 mM KCl could increase the lability of the histone-DNA contacts and promote translocation of a few bp of DNA. To test this possibility, we repeated the ensemble FRET

experiment at 70 mM KCl (*Figure 1—figure supplement 3D*). Under these conditions we did not observe a significant change in FRET explaining the absence of translocation in the 70 mM KCl reconstructions.

Unlike the structures at 70 mM KCl, the structures obtained at 140 mM KCl show a higher proportion of singly bound SNF2h at SHL-2 compared to SHL+2. Yet all the nucleosomes in the higher salt conditions (140 mM KCl) show the 2 bp DNA translocation in the same direction. This result seems paradoxical as a SNF2h protomer bound at SHL-2 is opposite to what is expected for the observed direction of DNA translocation (*Dang and Bartholomew, 2007*; *Kagalwala et al., 2004*; *Leonard and Narlikar, 2015*; *Schwanbeck et al., 2004*; *Zofall et al., 2006*). Comparison of the individual protomers in the doubly bound complex obtained at 70 mM KCl indicates that the SNF2h protomer at SHL+2 is conformationally more flexible than that at SHL-2 (*Figure 2A*, *Figure 2—figure supplement 2A*). This increased flexibility is consistent with the protomer at SHL+2 being the active protomer. We speculate that the increased dynamics of the SNF2h protomer bound at SHL+2 makes it more prone to dissociate during the cryo-EM grid preparation procedure carried out in 140 mM KCl. We therefore interpret the structures captured at 140 mM KCl as arising from partial disassembly of a doubly bound translocated complex, in which the protomer bound at SHL+2 has promoted translocation.

Based on the above comparisons, we conclude that the reconstructions obtained at 70 mM KCl, represent reaction intermediates that are poised to translocate by exploiting specific deformations in the octamer conformation, while the reconstructions obtained at 140 mM KCl represent translocated SNF2h-nucleosome states in which the deformed octamer has relaxed to its canonical conformation. We next use the 3.4 Å reconstruction of the translocated state to identify new SNF2h-nucleosome interactions and assess their role in nucleosome sliding.

## The role of SNF2h-nucleosome interactions in nucleosome sliding

Similar to observations for the Swi2/Snf2 ATPase domain and Chd1, our structure implies a large conformational rearrangement in the RecA lobes of SNF2h upon nucleosome binding (*Farnung et al., 2017*; *Hauk et al., 2010*; *Liu et al., 2017*; *Sundaramoorthy et al., 2018*; *Sundaramoorthy et al., 2017*; *Xia et al., 2016*; *Yan et al., 2016*) (*Figure 3B*, *Figure 3—figure supplement 2*). A large conformational change is also implied for the brace helix extending from lobe 2 (*Figure 3—figure supplement 3A*, orange helices) (*Liu et al., 2017*; *Yan et al., 2016*).

We see several interactions between RecA lobe two and the H4 tail (*Figure 3C*). Previous work has shown that the basic patch residues (KRHRK) of the H4 tail substantially activate ISWI remodeling, in part by counteracting the auto-inhibitory AutoN region (*Clapier et al., 2002*; *Clapier et al., 2001*; *Clapier and Cairns, 2012*; *Dang et al., 2006*; *Hamiche et al., 2001*; *Racki et al., 2014*). The crystal structure of *Myceliophthora thermophila* ISWI revealed that AutoN forms a structured domain that binds a cleft between the two RecA lobes (lobe 1 and lobe 2) of the ATPase domain (*Yan et al., 2016*). In our 3.4 Å structure we find that the AutoN domain has moved away from this cleft and one RecA lobe is flipped over the other to form a new cleft for nucleosome binding (*Figure 3B*, *Figure 3—figure supplement 3*). Most of AutoN is not resolved suggesting conformational flexibility in this state. Some residues of the RecA lobes that were shown to interact with AutoN in the unbound *Myceliophthora thermophila* ISWI structure now engage with the nucleosomal DNA and the H4 tail (*Yan et al., 2016*). Among these residues is an acidic surface on lobe two that directly engages the histone H4 tail basic patch (*Figure 3C*, top left). This exchange of interactions provides a structural explanation for how the H4 tail relieves autoinhibition by AutoN (*Clapier and Cairns, 2012*). Similar interactions were observed previously in the crystal structure of the isolated ATPase lobe 2 of *Myceliophthora thermophila* ISWI with an H4 tail peptide (*Yan et al., 2016*).

In addition to the H4 tail, the structure reveals new interactions between a loop within RecA lobe two and the globular region of histone H3 (*Figure 3C*, top right). Two residues K440 and K443 are in close proximity to make these direct contacts. Of these K443 is unique to and conserved across ISWI family members (*Figure 3—figure supplement 4*). Mutating K440 or K443 to alanine resulted in 4- and 6-fold defects respectively in the maximal rates of remodeling (*Figure 3D–F*). The K443A mutation had a much smaller effect on ATPase activity (≤2 fold), suggesting a role in the coupling of ATP hydrolysis to nucleosome sliding (*Figure 3—figure supplement 5*). Based on these functional effects we speculate that the interactions with H3 may help stabilize octamer deformation during SNF2h remodeling. Another residue in this loop, D442, is in proximity but not close enough to make

a direct contact. Mutating it had a small (2-fold) stimulatory effect on remodeling, suggesting this residue plays a modest autoinhibitory role (*Figure 3D–F*). While comparable H3 contacts by SWI/SNF motors have not been observed, similar functionally important interactions between the H3 alpha one helix and an insertion in RecA lobe two are found in structures of the Chd1-nucleosome complex (*Sundaramoorthy et al., 2018*).

In terms of DNA interactions, several residues of the conserved lobe 1 and lobe two motifs are in proximity to the DNA at the SHL-2 region (*Figure 4A*). Two residues conserved between ISWI and SWI/SNF ATPases are W581 and N448 (*Figure 3—figure supplement 4*). While mutation of W581 in the *S. cerevisiae* Swi2/Snf2 ATPase domain causes a > 10 fold defect in remodeling, mutating this residue within SNF2h caused a 3-fold defect in the maximal rate of nucleosome sliding (*Figure 4B–D*) (*Liu et al., 2017*). Mutating N448 causes a ~ 9 fold defect in the maximal rate of remodeling ($k_{max}$, *Figure 4B–D*) without causing a significant defect in the $K_m$ (data not shown), suggesting that this interaction makes a bigger contribution during catalysis rather than binding. In contrast, both mutations had smaller effects on DNA-stimulated ATPase activity ($\leq 2$ fold increase, *Figure 3—figure supplement 5*), suggesting these residues contribute to coupling of ATP hydrolysis to DNA translocation. A distortion from the normal path of nucleosomal DNA is also observed at SHL-2 (*Figure 4E*). This distortion is consistent with previous biochemical observations that the DNA path is altered at the site where the ATPase domain binds (*Schwanbeck et al., 2004*; *Zofall et al., 2006*).

For *S. cerevisiae* Swi2/Snf2, a positively charged patch in RecA lobe one is positioned to bind on the DNA gyre below SHL±2 near SHL ± 6 (*Liu et al., 2017*). Charge reversal point mutations in this patch modestly reduced remodeling by Swi2/Snf2 (~2 fold), while multiple charge reversal mutations had severe defects (*Liu et al., 2017*). These cross-gyre contacts are not conserved with CHD family remodelers, which instead have this region in RecA lobe one replaced with acidic residues. This results in lobe one being positioned farther away from SHL ± 6 than in Swi2/Snf2 (*Sundaramoorthy et al., 2018*). In our 3.4 Å structure, we see RecA lobe one positioned near SHL +6 but without the close contacts seen for Swi2/Snf2 (*Figure 4A*). Mutation of a lysine residue that is proximal to SHL+6 to glutamate (K298E), resulted in only a ~ 3 fold reduction in maximal sliding activity (*Figure 4B–D*) suggesting a modest role for this interaction in remodeling by SNF2h.

## Discussion

The histone octamer is often conceptualized as a steric barrier to accessing DNA. In this context ISWI motors have the difficult task of sliding nucleosomes despite the constraints imposed by the histone octamer. Yet these motors are able to cause rapid and directional nucleosome translocation without disassembling the octamer (*Zhou et al., 2016*). Our results here suggest that instead of acting as a barrier that needs to be overcome, the histone octamer may actively participate in the reaction by acting as a deformable medium for allosteric control. Below we discuss the mechanistic significance of these new findings in the context of previous discoveries.

We have previously found that SNF2h binding in the presence of ADP-BeF$_x$ increases the dynamics of a subset of buried histone residues (*Sinha et al., 2017*). These dynamics play a role in the initiation of nucleosome sliding from SHL ± 2 and the directionality of sliding. Relevant to these observations, several prior studies have provided evidence for intrinsic dynamics within nucleosomes. These include spontaneous unpeeling of nucleosomal DNA (*Li et al., 2005*), the identification by EM of alternative configurations of histone helices within distorted nucleosomes (*Bilokapic et al., 2018b*) and the identification of histone mutants than increase the spontaneous dynamics (*Kitevski-LeBlanc et al., 2018*) of core histone resides. It has been further demonstrated that core histone dynamics enable spontaneous nucleosome sliding (*Bilokapic et al., 2018a*). However the scale and nature of the octamer deformations promoted by a remodeler such as SNF2h has been unknown. The EM reconstructions obtained at 70 mM KCl (*Figure 2*) provide a starting point for understanding the structural basis of SNF2h mediated octamer fluctuations. Two unexpected features stand out. First, unlike the defined alternative helical conformations observed previously in nucleosomes alone, the EM density changes that we observe imply an ensemble of states with increased disorder in the histone conformation (*Bilokapic et al., 2018b*; *Bilokapic et al., 2018a*). Second, the increased disorder is asymmetric and distal from the bound SNF2h protomer suggesting an allosteric mode of octamer deformation.

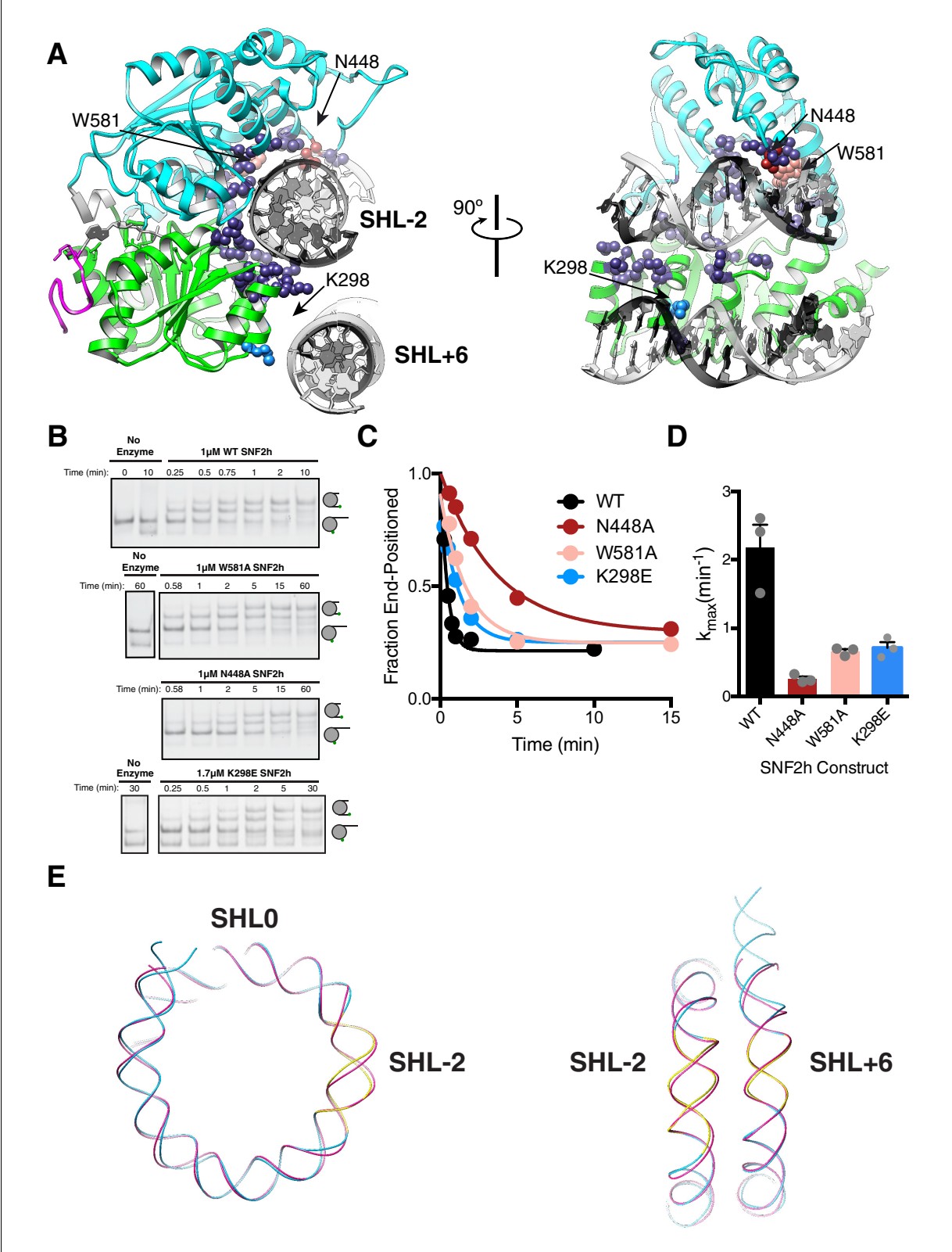

**Figure 4.** DNA contacts in the SNF2h-Nucleosome structure. (**A**) SNF2h residues contacting the nucleosome are shown in spheres. In light blue is a contact with the second gyre near SHL + 6. In red, pink, and light blue are residues mutated in this study. (**B**) Native gel remodeling assay of DNA contact mutants. (**C**) Quantification of gels in (**B**). The x-axis is zoomed to show detail. The un-zoomed plots are in *Figure 3—figure supplement 6*. (**D**)

*Figure 4 continued on next page*

*Figure 4 continued*
Mean and standard error of rate constants derived from three experiments. (**E**) Comparison of the nucleosomal DNA in the SNF2h-nucleosome structure (blue) with the unbound structure (magenta) (PDBID: 1K × 5). Contacts with the remodeler at SHL-2 and SHL+6 are highlighted in yellow.
DOI: https://doi.org/10.7554/eLife.46057.023
The following source data is available for figure 4:

**Source data 1.** Values plotted in C and D.
DOI: https://doi.org/10.7554/eLife.46057.024

We propose that the structure captured at lower salt mimics a reaction intermediate in which SNF2h action results in asymmetric disordering of specific histone regions prior to translocation (*Figure 5*). The apparent asymmetric nature of the octamer deformation could help explain a long-standing question about SNF2h protomer coordination. SNF2h assembles on nucleosomes as a face-to face dimer, but what prevents the two protomers from simultaneously translocating DNA in opposite directions, resulting in a stalled nucleosome? Our findings raise the possibility that the active protomer prevents a tug of war by allosterically altering the acidic patch surface required by the second protomer. In addition, the disorder near the dyad is expected to promote the directional translocation of DNA initiated by the active protomer. We further propose that collapse of the reaction intermediate results in an elemental translocation step, which is mimicked by the structure captured at higher salt (*Figure 5*). Consistent with this possibility, previous single-molecule studies have indicated that the elemental translocation event driven by ISWI enzymes is 1–2 bp (*Deindl et al., 2013*).

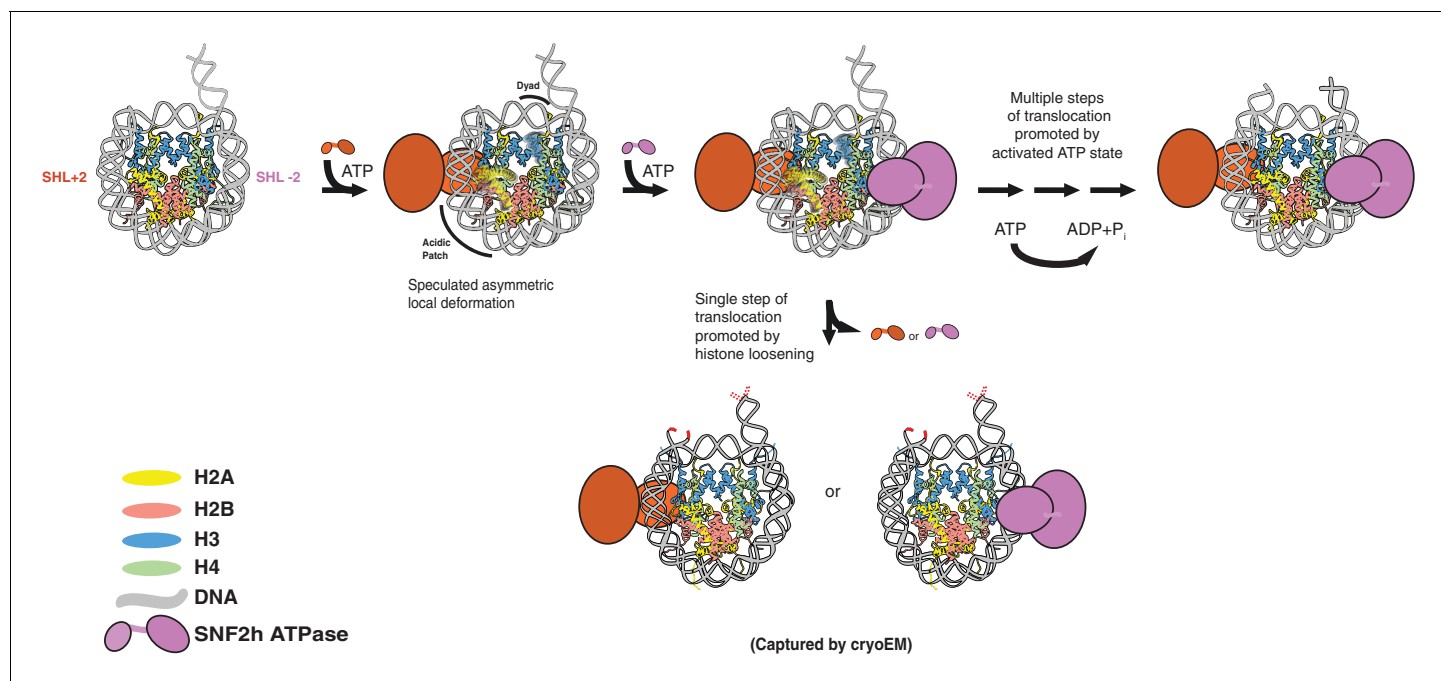

**Figure 5.** Speculative model that places SNF2h-nucleosome Cryo-EM structures within SNF2h reaction cycle . Two protomers of SNF2h bind to the nucleosome along with ATP. Based on previous work, the directionality of nucleosome sliding is determined by the motor that engages the longer flanking DNA (*Leonard and Narlikar, 2015*). By this model, the SNF2h motor bound at SHL+2 (orange protomer) will be the active motor and determine the direction of sliding because it would contact the 60 bp flanking DNA with its HSS domain (*Leonard and Narlikar, 2015*). For simplicity, the HSS domain is not shown. Binding of the SHL+2 protomer asymmetrically deforms the acidic patch and histone H3 near the dyad on the opposite face of the histone octamer. The second protomer can bind at SHL-2, but cannot act because deformation of the acidic patch inhibits its ability to slide nucleosomes. The SNF2h complex with the deformed octamer represents an intermediate that is poised for translocation. Processive DNA translocation is enabled by successive ATP hydrolysis cycles from this activated intermediate, moving DNA in 1–2 bp fundamental increments. We speculate that the cryo-EM structure captured at low salt represents the deformed intermediate, while the structure captured at high salt represents a collapsed product state in which the nucleosome is translocated by 2 bp (translocated bases are highlighted in red).
DOI: https://doi.org/10.7554/eLife.46057.025

Future studies looking at structures in additional nucleotide states will help advance our understanding of how changes in ISWI conformation are coupled to changes in octamer dynamics. This will allow for a mechanistic understanding of chromatin remodelers that parallels that of molecular motors such as kinesin and myosin.

# Materials and methods

**Key resources table**

| Reagent type (species) or resource | Designation | Source or reference | Identifiers | Additional information |
|---|---|---|---|---|
| Strain, strain background (*Escherichia coli*) | Rosetta (DE3) | Millipore sigma | 70954 | Chemically competent cells |
| Strain, strain background (*Escherichia coli*) | BL1 (DE3) pLysS | Agilent Technologies | 200132 | Chemically competent cells |
| Recombinant DNA reagent | core Widom 601 (bold) andflanking DNAsequences | (*Lowary and Widom, 1998*) | | 5'-CGGCCGCCCTGGAGAATCCCG GTGCCGAGGCCGCTCAATTGGTC GTAGACAGCTCTAGCACCGCTTA AACGCACGTACGCGCTGTCCCCC GCGTTTTAACCGCCAAGGGGATT ACTCCCTAGTCTCCAGGCACGTGT CAGATATATACATCCTGTGCATGT ATTGAACAGCGACCTTGCCGGTGC CAGTCGGATAGTGTTCCGAGCTCC CACTCTAGAGGATCCCCGGGTACC-3' |
| Recombinant DNA reagent | 601 plasmid | (*Lowary and Widom, 1998*) | | PCR template |
| Recombinant DNA reagent | pBH4-SNF2h | (*Leonard and Narlikar, 2015*) | | Expression plasmid |
| Recombinant DNA reagent | Pet3a-H2A | (*Yang et al., 2006*) | | Expression plasmid |
| Recombinant DNA reagent | Pet3a-H2B | (*Yang et al., 2006*) | | Expression plasmid |
| Recombinant DNA reagent | Pet3a-H3 | (*Yang et al., 2006*) | | Expression plasmid |
| Recombinant DNA reagent | Pet3a-H4 | (*Yang et al., 2006*) | | Expression plasmid |
| Sequence-based reagent | 601 core forward primer | IDT | | 5'-CTGGAGAATCCCGGTGCCG-3' |
| Sequence-based reagent | 601 + 60 reverse primer | IDT | | 5'-AGAGTGGGAGCTCGG AACAC-3' |
| Sequence-based reagent | Cy3- 601 core forward primer | IDT | | 5'-/Cy3/CTGGAGAATCCCGGTGCCG-3' |
| Sequence-based reagent | Cy5- 601–9 forward primer | TriLink Biotechnologies | | 5'-{Cyanine5-C6-NH}GCG GCC GCCCTGGAGAATCC-3' |
| Sequence-based reagent | Bio- 601 + 78 reverse primer | IDT | | 5'-/5BioTeg/GGTACCCGG GGA TCCTCTAGAG-3' |
| Sequence-based reagent | 601–120F C149cy5 | Iba | | 5'-GGCACGTGTCAGATATATA CATCCTGTG5ATGTATTGAACA-3' 5 = cy5 C6-Amino-2'deoxycytidine |
| Sequence-based reagent | SNF2h K298E forward primer | IDT | | 5'-GAGAAGTCTGTGTTCGA AAAATTTAATTGGAG-3' |

*Continued on next page*

*Continued*

| Reagent type (species) or resource | Designation | Source or reference | Identifiers | Additional information |
|---|---|---|---|---|
| Sequence-based reagent | SNF2h K298E reverse primer | IDT | | 5'-CTCCAATTAAATTTTT CGAAC ACAGACTTCTC-3' |
| Sequence-based reagent | SNF2h K440A forward primer | IDT | | 5'-CTCAACTCAGCAGGCG CGATGGACAAAATGAGG-3' |
| Sequence-based reagent | SNF2h K440A reverse primer | IDT | | 5'-CCTCATTTTGTCCATCG CGCCTGCTGAGTTGAG-3' |
| Sequence-based reagent | SNF2h D442A forward primer | IDT | | 5'-CTCAGCAGGCAAGATGGCG AAAATGAGGTTATTGAAC-3' |
| Sequence-based reagent | SNF2h D442A reverse primer | IDT | | 5'-GTTCAATAACCTCATTTTCG CCATCTTGCCTGCTGAG-3' |
| Sequence-based reagent | SNF2h K443A forward primer | IDT | | 5'-CAGCAGGCAAGATGGACGC GATGAGGTTATTGAACATC-3' |
| Sequence-based reagent | SNF2h K443A reverse primer | IDT | | 5'-GATGTTCAATAACCTCATCG CGTCCATCTTGCCTGCTG-3' |
| Sequence-based reagent | SNF2h N448A forward primer | IDT | | 5'-GACAAAATGAGGTTATTGG CGATCCTAATGCAGTTGAG-3' |
| Sequence-based reagent | SNF2h N448A Reverse primer | IDT | | 5'-CTCAACTGCATTAGGATCG CCAATAACCTCATTTTGTC-3' |
| Sequence-based reagent | SNF2h W581A forward primer | IDT | | 5'-GTAATTTTGTATGATTCTGA TGCGAATCCCCAAGTAGATCTTC-3' |
| Sequence-based reagent | SNF2h W581A reverse primer | IDT | | 5'-GAAGATCTACTTGGGGATTCG CATCAGAATCATACAAAATTAC-3' |
| Peptide, recombinant protein (*Homo sapiens*) | SNF2h | (*Leonard and Narlikar, 2015*) | | |
| Peptide, recombinant protein (*Xenopus laevis*) | Histone H2A | (*Luger et al., 1997*) | | |
| Peptide, recombinant protein (*Xenopus laevis*) | Histone H2B | (*Luger et al., 1997*) | | |
| Peptide, recombinant protein (*Xenopus laevis*) | Histone H3 | (*Luger et al., 1997*) | | |
| Peptide, recombinant protein (*Xenopus laevis*) | Histone H4 | (*Luger et al., 1997*) | | |
| Peptide, recombinant protein (*Xenopus laevis*) | Histone H3 33C | (*Rowe and Narlikar, 2010*) | | |
| Peptide, recombinant protein (*Escherichia virus T4*) | T4 DNA Ligase | New England Biolabs | Cat #: M0202L | |
| Peptide, recombinant protein (*Bos taurus*) | Catalase | Sigma | Cat #: E3289 | |

*Continued on next page*

Continued

| Reagent type (species) or resource | Designation | Source or reference | Identifiers | Additional information |
|---|---|---|---|---|
| Peptide, recombinant protein (*Aspergillus niger*) | Glucose oxidase | Sigma | Cat #: G2133 | |
| Peptide, recombinant protein (*Oryctolagus cuniculus*) | Lactate dehydrogenase | Sigma | Cat #: 427217 | |
| Peptide, recombinant protein (*Oryctolagus cuniculus*) | Pyruvate Kinase | Sigma | Cat #: 10128155001 | |
| Chemical compound, drug | ATP | GE | Cat #: 27-2056-01 | |
| Chemical compound, drug | $\gamma$-$^{32}$P-ATP | Perkin Elmer | Cat #: Blu002Z250uC | |
| Chemical compound, drug | ADP | Millipore sigma | Cat #: 117105 | |
| Chemical compound, drug | Cy3-maleimide | Lumiprobe | Cat #: 21080 | |
| Chemical compound, drug | Cy5-maleimide | Lumiprobe | Cat #: 43080 | |
| Chemical compound, drug | dNTPs | Allstar Scientific | Cat #: 471-5DN | |
| Chemical compound, drug | N-(2-aminoethyl)—3-aminopropyl trimethoxysilane | United Chemicals | Cat #: A0700 | |
| Chemical compound, drug | mPEG-SVA | Laysan Bio | | |
| Chemical compound, drug | biotin-PEG-SVA | Laysan Bio | | |
| Chemical compound, drug | acetylated BSA | Promega | Cat #: R3691 | |
| Chemical compound, drug | Neutravidin | Life Technologies | A2666 | |
| Chemical compound, drug | Trolox | Sigma | Cat #: 238813 | |
| Chemical compound, drug | 10XTBE | Bio-Rad | Cat #: 161–0770 | |
| Chemical compound, drug | Acrylamide/ Bis-acrylamide | Bio-rad | Cat #: 161–0146 | |
| Chemical compound, drug | HEPES | Fisher | Cat #: BP310 | |
| Chemical compound, drug | Tris Base | Thermo Fisher | Cat #: BP1525 | |
| Chemical compound, drug | NaCl | RPI | Cat #: S23020 | |
| Chemical compound, drug | KCl | Sigma | Cat #: P3911 | |
| Chemical compound, drug | $MgCl_2$ | RPI | Cat #: M24000 | |
| Chemical compound, drug | Glycerol | Sigma | Cat #: G7893 | |

*Continued*

| Reagent type (species) or resource | Designation | Source or reference | Identifiers | Additional information |
|---|---|---|---|---|
| Chemical compound, drug | NP40 (IGEPAL) | Sigma | Cat #: I8896 | |
| Chemical compound, drug | 2-Mercaptoethanol | Sigma | Cat #: M3148 | |
| Chemical compound, drug | Glucose | RPI | Cat#: G32045 | |
| Chemical compound, drug | NADH | Millipore Sigma | Cat#: 481913 | |
| Chemical compound, drug | Phosphoenol Pyruvate | Thermo Fisher | Cat#: NC9842221 | |
| Software, algorithm | Prism 6 | Graphpad | | |
| Software, algorithm | Traces | https://github.com/stephlj/Traces | | |
| Software, algorithm | pyhsmm | https://github.com/mattjj/pyhsmm | | |
| Software, algorithm | Slopey | https://github.com/stephlj/slopey | | |
| Software, algorithm | PyEM | https://github.com/asarnow/pyem | | |
| Software, algorithm | Gautomatch | http://www.mrc-lmb.cam.ac.uk/kzhang/ | | |
| Software, algorithm | SerialEM | (*Mastronarde, 2005*) | | |
| Software, algorithm | RELION 3.0 | (*Zivanov et al., 2018*) | | |
| Software, algorithm | Motioncor2 | (*Zheng et al., 2017*) | | |
| Software, algorithm | GCTF | (*Zhang, 2016*) | | |
| Software, algorithm | UCSFImage4 | (*Li et al., 2015*) | | |
| Software, algorithm | EMAN2 | (*Tang et al., 2007*) | | |
| Software, algorithm | CryoSPARC | (*Punjani et al., 2017*) | | |
| Software, algorithm | Diffmap.exe | http://grigorefflab.janelia.org/diffmap | | |
| Software, algorithm | Coot | (*Emsley et al., 2010*) | | |
| Software, algorithm | Phenix | (*Adams et al., 2010*) | | |
| Software, algorithm | ImageJ | https://imagej.nih.gov/ij/ | | |
| Other | Superdex 200 increase 10/300 GL | GE | Cat. #: 29091596 | |
| Other | HiTrap QXL column | GE | Cat. #: 17-5159-01 | |
| Other | Superdex 200 HiLoad 26/600 | GE | Cat. #: 28989336 | |
| Other | TALON metal affinity resin | Clontech | Cat. # 635503 | |

## Protein expression, purification and complex preparation for cryo-EM

Human SNF2h was expressed in *Escherichia coli* BL21(DE3) Rosetta cells and purified as previously described (*Leonard and Narlikar, 2015*). SNF2h mutations were generated by site directed mutagenesis using the quick change protocol (Stratagene). Recombinant *Xenopus laevis* histones were expressed in *E. coli*, purified from inclusion bodies and assembled into octamers as described previously (*Luger et al., 1999*). Briefly, histone protein octamer was reconstituted from denatured purified histones via refolding in high salt buffer and purified on a Superdex 200 increase 10/300 GL size exclusion column (GE Healthcare). DNA containing the Widom 601 positioning sequence with 60 bp of flanking DNA was made by large scale PCR with Taq DNA polymerase and purified by native PAGE as described previously (*Zhou and Narlikar, 2016*). For remodeling assays, 5' Cy3 labeled DNA was made by large scale PCR with a primer labeled at the 5' end with the fluorophore (*Zhou and Narlikar, 2016*). DNA labeled at two locations for ensemble FRET experiments was prepared by large scale PCR of two separate DNA templates (207 bp DNA 5'-cy3 labeled, using labeled primers (IBA Life Sciences). 120 µg of each DNA template was digested with AflIII at 37°C overnight and purified by native PAGE. Purified DNA fragments were then ligated with 8000 units of T4 DNA ligase with 1 mM ATP•MgCl$_2$ (New England Biolabs) for 20 min at room temperature and purified again by native PAGE. Nucleosomes were reconstituted by the salt gradient dialysis method and purified by glycerol gradient centrifugation (*Zhou and Narlikar, 2016*). Purified nucleosomes were flash frozen in liquid nitrogen and stored at −80°C.

The sequence of the 601 sequence with 60 bp of flanking DNA (207 bp DNA) is as follows:
CTGGAGAATCCCGGTGCCGAGGCCGCTCAATTGGTCGTAGACAGCTCTAGCACCGC TTAAACGCACGTACGCGCTGTCCCCCGCGTTTTAACCGCCAAGGGGATTACTCCCTAGTC TCCAGGCACGTGTCAGATATATACATCCTGTGCATGTATTGAACAGCGACCTTGCCGGTGCCAG TCGGATAGTGTTCCGAGCTCCCACTCT

Pre-assembled nucleosomes were first dialyzed overnight into 25 mM HEPES pH 7.5 to remove glycerol before sample preparations for cryo-EM. To prepare the nucleosome-SNF2h complex, nucleosomes were mixed with purified SNF2h on ice, then incubated at room temperature for 10 min before applying to the grids for plunge freezing. Initially, the complex was prepared by mixing 1.45 µM 0/60 nucleosomes with 5 µM SNF2h in 20 mM HEPES pH 7.5, 70 mM KCl, 0.5 mM ADP-Mg$^{2+}$, 0.5 mM MgCl$_2$, 0.5 mM BeF$_x$ (1:5 BeCl$_2$:NaF). This sample yielded a reconstruction of nucleosome with two SNF2h bound at 8.4 Å resolution by using a scintillator-based camera. The other sample was prepared by mixing 0.625 µM 0/60 nucleosomes with 1.25 µM SNF2h in 0.5 mM ADP-Mg, 0.5 mM BeF$_x$, 0.5 mM MgCl$_2$, 140 mM KCl, 3 mM Tris pH 7.5, 1.25 mM HEPES pH 7.5, 1.5% glycerol. This resulted in high-resolution reconstructions using direct electron detection camera. While the resolution difference is mainly caused by the camera technology and unlikely to be related to the differences in sample preparation, the salt concentration could be amplified during the process of blotting and plunge freezing. For the nucleosome alone reconstruction, 1 µM nucleosomes were used directly after the dialysis.

When plunge freezing cryo-EM grids, both protein and solutes in buffer are concentrated but to very different extents. The concentration of SNF2h-nucleosome samples on cryo-EM grids are roughly ~100 µM, estimated from the number of particles seen in micrographs. This concentration increase is mostly caused by the volume reduction (~10,000 fold) during blotting, in which filter papers preferably absorb water with chemical solutes over proteins. It is not possible to estimate the final concentration of KCl and ADP-BeF$_x$ in a frozen cryo-EM grid, but it may also increase several folds from the sample applied to EM grids, because of evaporation between blotting and freezing. Thus, 70 mM KCl could become as high as ~200 mM and 140 mM KCl become ~400 mM before freezing. If so, it would profoundly impact the binding affinity of SNF2h to the nucleosome, and/or the affinity of ADP-BeF$_x$ for SNF2h. We also speculate that macromolecular interactions that are destabilized by the higher ionic strength are more susceptible to being disassembled by the air-water interface during plunge freezing.

## Negative stain EM

Grids of negatively stained sample were prepared as described (*Ohi et al., 2004*). Sample from these grids was then observed on a Tecnai T12 (FEI) operated at 120kV (*Figure 2—figure supplement 3A–D*). From collected micrographs, monodisperse particles were picked manually, windowed

out and subjected to template-free 2D classification using RELION (*Scheres, 2012*). Selected representative classes are show in *Figure 2—figure supplement 3E*. Only monodispersed particles were pick for further processing.

## Cryo-EM data acquisition

Cryo-EM grids of nucleosome-SNF2h or nucleosome alone samples were prepared following established protocol (*Liao et al., 2013*). Specifically, 2.5 µl of nucleosome-SNF2h complexes (or 3 µl of samples of nucleosome alone) were applied to a glow discharged Quantifoil holey carbon grid (1.2 µm hole size, 400 mesh), blotted in a Vitrobot Mark I (FEI Company) using 6 s blotting at 100% humidity, and then plunge-frozen in liquid ethane cooled by liquid nitrogen.

The scintillator-based camera dataset was collected at liquid nitrogen temperature on a Tecnai TF20 (Thermo Fisher Scientific) electron microscope equipped with field emission gun (FEG) electron source and operated at 200kV. Images were recorded on a TemF816 8k × 8 k CMOS camera (TVIPS GmbH) at a nominal magnification of 62,000X, corresponding to a pixel size of 1.2 Å/pixel on the specimen, with a defocus in the range from 1.8 to 2.9 µm. Data collection follows the low-dose procedure using UCSFImage4 (*Li et al., 2015*). The K2 camera dataset of nucleosome-SNF2h complex was collected using UCSFImage4 on a TF30 Polara electron microscope (Thermo Fisher Scientific) equipped with a FEG source and operated at 300 kV. Specifically, images were recorded in super resolution counting mode using a K2 Summit direct electron detection camera (Gatan Inc) at a nominal magnification of 31,000X, corresponding to a calibrated physical pixel size of 1.22 Å/pixel. The dose rate on camera was set to 8.2 counts (corresponding to 9.9 electrons) per physical pixel per second. The total exposure time was 6 s, leading to a total accumulated dose of 41 electrons per $Å^2$ on the specimen. Each image was fractionated into 30 subframes, each with an accumulated exposure time of 0.2 s. Images were recorded with a defocus in the range from 1.5 to 3.0 um. The K2 dataset of nucleosome alone was collected in the same microscope under the identical imaging conditions, except SerialEM was used for automated acquisition (*Mastronarde, 2005*).

## Image processing

For dataset collected with scintillator based camera, there is no movie stack related image processing, such as motion correction and dose weighting. Otherwise, same software packages and procedures were used as for the K2 datasets.

For K2 datasets, movie stacks were corrected for both global and local motions using Motion-Cor2 v1.0.0., which outputs both dose-weighted and un-weighted sum of corrected subframes (*Zheng et al., 2017*). The output images were first visually inspected for particle distribution. The non-dose-weighted images were used for CTF parameter determination using GCTF (*Zhang, 2016*). The estimated image resolution and quality of Thon ring fitting were inspected manually and images with poor quality were removed from further image processing. For the rest of image processing, only dose-weighted sums were used. For particle picking, an initial ~1000 particles were manually picked using e2boxer (EMAN2) (*Tang et al., 2007*), followed by two-dimensional (2D) reference free alignment and classification by using Relion2 (*Scheres, 2012*). Six unique 2D class averages were used as template reference for an automated particle picking using Gautomatch (http://www.mrc-lmb.cam.ac.uk/kzhang/). Only monodispersed particles are picked. Particle aggregations, which are occasionally seen, are excluded. All picked particles were subject to reference free 2D classifications. Particles within 2D classes that show clear nucleosome features were selected and were further inspected visually to remove any remaining 'junk' particles. The total number of particles in each dataset is listed in *Figure 2—figure supplement 2* and *Tables 1–2*. For initial model generation and subsequent runs, cryoSPARC 'Ab initio', 'Homogeneous refinement' and 'Heterogeneous refinement' procedures were employed (*Punjani et al., 2017*). For masked refinement and classification without alignment, we used RELION 2. The detailed scheme of classifications and refinements are shown in the *Figure 2—figure supplement 2*. The refinement follows gold-standard refinement procedure (*Scheres and Chen, 2012*), and the final resolutions were estimated using Fourier Shell Correlation (FSC) equals 0.143 criterion (*Rosenthal and Henderson, 2003*).

The K2 dataset was used in two independent analyses. In the first analysis, we picked 120,533 particles and classified into 50 2D class averages. Particles within low quality 2D classes were removed. Through sequential classification and refinement, we obtained a 3.8 Å reconstruction of

**Table 1.** Summary table for data collection and refinement on TVIPS 816 scintillator-based camera

| Dataset | Single-bound SNF2h-nucleosome complex at SHL + 2 | Doubly-bound SNF2h-nucleosome at SHL ± 2 |
|---|---|---|
| Microscope | TF20 (FEI) | TF20 (FEI) |
| Voltage (kV) | 200 | 200 |
| Camera | TemF816 8k × 8 k CMOS (TVIPS) | TemF816 8k × 8 k CMOS (TVIPS) |
| Magnification | 62.000 | 62.000 |
| Pixel size (Å) | 1.2 | 1.2 |
| Defocus range (μm) | −1.8: −2.9 | −1.8: −2.9 |
| Number of images | 766 | 766 |
| Total electron dose (e$^-$/Å$^2$) | 25 | 25 |
| Number of frames | - | - |
| Initial number of particles | 450322 | 450322 |
| Particles selected after 2D cleanup | 379540 | 379540 |
| Particles in final reconstruction | 32233 | 57060 |
| Final resolution (Å) | 8.4 | 8.4 |

DOI: https://doi.org/10.7554/eLife.46057.026

single SNF2h-bound nucleosome. To separate particles with SNF2h bound to SHL+2 and SHL-2, we first used volume subtraction procedure to subtract the SNF2h from all particles followed by unambiguously aligning all nucleosome particles using the flanking DNA as a fiducial mark. We then performed a focused classification on SNF2h, which lead to two main subclasses: one reconstruction at 3.9 Å resolution has SNF2h bound to the SHL-2 position. The other one at 6.9 Å resolution has SNF2h bound to the SHL + 2 position (*Figure 1—figure supplement 2*). Both reconstructions show a 2 bp translocation of DNA (*Figure 1—figure supplement 3*).

Independently, we re-processed frame motion correction by using MotionCor2 v1.2.1, with option accounting for in-frame motion enabled (InFmMotion 1). The motion corrected images were

**Table 2.** Summary table for data collection and refinement on Gatan K2-Summit direct electron detector camera.

| Dataset | Single-bound SNF2h-nucleosome complex: SHL-2/SHL + 2 | Single-bound SNF2h-nucleosome complex, SHL-2 | Core nucleosome |
|---|---|---|---|
| Microscope | TF30 (FEI) | TF30 (FEI) | TF30 (FEI) |
| Voltage (kV) | 300 | 300 | 300 |
| Camera | K2 summit (Gatan) | K2 summit (Gatan) | K2 summit (Gatan) |
| Magnification | 31.000 | 31.000 | 31.000 |
| Pixel size (Å) | 1.22 | 1.22 | 1.22 |
| Defocus range (μm) | −1.5: −3.0 | −1.5: −3.0 | −1.5: −3.0 |
| Number of images | 720 | 720 | 720 |
| Total electron dose (e$^-$/Å$^2$) | 42 | 42 | 42 |
| Number of frames | 30 | 30 | 30 |
| Initial number of particles | 120533 | 333430 | 24993 |
| Particles selected after 2D cleanup | 95879 | no cleanup | 19363 |
| Particles in final reconstruction | 27513/6241 | 43165 | 12130 |
| Final resolution (Å) | 3.9/6.9 | 3.4 | 7.4 |

DOI: https://doi.org/10.7554/eLife.46057.027

subjected again to independent particle picking and CTF estimation. In this reprocess, we started with 333,430 initially picked particles (*Figure 2—figure supplement 2C*). We skipped 2D classification for cleaning up the dataset, instead using 3D classification for this purpose. Using cryoSPARC v2 'Ab initio reconstruction' and 'Heterogeneous refinement', in subsequent rounds of subsorting, the final reconstruction contains 43,165 particles, which yielded 3.4 Å reconstruction using 'Non-homogeneous refinement'. We also attempted to separate particles into SNF2h bound to SHL+2 and SHL-2 positions using focused classification and signal subtraction. We succeeded by obtaining a 3.6 Å SHL-2 reconstruction from 28500 particles and a 4.5 Å reconstruction from 14665 particles. The latter likely still contains a mix of SHL+2 and SHL-2 that we were unable to separate further.

## Validation of 2 base-pair translocated nucleosome

We ruled out the possibility of nucleosome mis-assembly as described in the main text (*Figure 1—figure supplements 5–6*). Furthermore, the following additional experiments were carried out to rule out the possibility of any computational artifacts, such as incomplete separation of particles with SNF2h bound to SHL+2 and SHL-2 positions. In addition to the fact that the 3.9 Å reconstruction shows that DNA has a sharp ending without any weak extension, we calculated 20 bootstrapped 3D reconstructions using a subset of particles bootstrapped from the particles that were used to calculate the 3.9 Å map (*Figure 1—figure supplement 6*). These bootstrapped reconstructions show no significant variance at the location of this extra density. These observations demonstrate that this extra density is statistically significant and well defined, and cannot be contributed by misaligned particles. Furthermore, we calculated difference maps between all experimental maps determined from datasets recorded with K2-Summit and the map simulated from the atomic model of nucleosome without the two extra base-pairs at the exit side (*Figure 1—figure supplement 5*). We also calculated difference maps between the reconstruction of nucleosome alone and the reconstructions of SNF2h bound to either SHL+2 or SHL-2 (*Figure 1—figure supplement 5A–B*). These difference maps confirm the existence of extra DNA density at the exit side of our SNF2h-nucleosome complex reconstructions.

To calculate the variance map, we bootstrapped 5000 particles 20 times from the 3.9 Å SNF2h dataset and backprojected particles within these subsets to produce 20 reconstructions using relion_reconstruct in Relion2. The variance map between all reconstructions was then calculated. All scripts are included in PyEM (https://github.com/asarnow/pyem). The difference maps were calculated using program diffmap.exe (http://grigoriefflab.janelia.org/diffmap).

## Model building and refinement

For the nucleosome 0/60 nucleosome, we used Widom 601 structure crystal structure (*Vasudevan et al., 2010*) (PDBID: 3LZ1) and mutated the DNA to reflect the exact sequence used for the sample. We then used Coot (*Emsley et al., 2010*) and Phenix (phenix.realspacerefine) (*Adams et al., 2010*) to extend and fit the DNA into our structures, as well as ensure the correctness of the structure.

SNF2h was constructed using homology modeling, based on an ISWI crystal structure from *Myceliophthora thermophile* (PDBID: 5JXR) (*Yan et al., 2016*). We separated each of the domains and used rigid body fitting into the EM density. Subsequently, Coot and Phenix were used to adjust and modify parts of the model. For model cross-validation, the final structure was initially subjected to 0.1 Å random displacement and then refined against one of the two half-maps using Phenix. Subsequently, the refined pdb was converted to a density map and FSC curves were calculated between three maps: half map 1 (the refinement map, 'work'), half map 2 (not used for refinement, 'free') and the summed map. A very small difference between the 'work' and 'free' FSC curves indicates little-to-no effect of over-fitting of the atomic model. Figures were prepared using UCSF Chimera (*Pettersen et al., 2004*).

## Native gel remodeling assay

All remodeling reactions were performed under single turnover conditions (enzyme in excess of nucleosomes) using similar methods as described previously (*Zhou and Narlikar, 2016*). Reactions with SNF2h were performed at 20°C with 15 nM cy3-labeled nucleosomes, 12.5 mM HEPES pH 7.5, 2 mM Tris pH 7.5, 70 mM KCl, 5 mM ATP-MgCl$_2$, 3 mM MgCl$_2$,0.02% NP40, and ~3%(v/v) glycerol.

5 µL time points were quenched with equal volumes of stop buffer containing an excess of ADP and plasmid DNA. Nucleosomes were resolved on a 6% polyacrylamide 0.5X TBE native gel. Reactions were visualized by scanning on a Typhoon variable mode imager (GE Healthcare) and quantified using ImageJ.

## Ensemble FRET assay

Steady state fluorescence measurements were performed on an ISS K2 fluorometer equipped with a 550 nm short pass and 535 nm long pass filter in front of excitation and emission monochromators respectively. Fluorescence emission spectra were collected by excitation at 515 nm and emission intensities measured between 550–750 nm in 5 nm wavelength increments. FRET efficiency was determined by the following equation:

$$\text{FRET Efficiency} = \frac{Em_{665}}{Em_{665} + Em_{565}}$$

Where $Em_{665}$ and $Em_{565}$ are the maximal acceptor and donor emission intensities at 665 nm and 565 nm respectively. Reactions were carried out with a final volume of 80 µL and with final concentrations of 8 nM labeled nucleosomes, 12.5 mM HEPES pH 7.5, 2 mM Tris pH 7.5, 0.5 mM $MgCl_2$, 0.02% NP40,~4% glycerol at 20˚C. Each reaction was incubated for ~10 min before an initial emission spectrum was obtained. Reactions were then initiated with 3 µL 2 µM of SNF2h and 0.5 mM ADP-$BeF_x$-$MgCl_2$ (final concentration) or buffer and emission spectra were obtained at various time points after initiating the reactions. Kinetic measurements were normalized to the FRET efficiency of the initial measurement.

## Single molecule FRET

Experiments were performed as in *Gamarra et al. (2018)*, except that the imaging buffer was 53 mM HEPES-KOH, pH 7.5 at 22˚C, 9.1 mM Tris-acetate, pH 7.5 at 22˚C, 140 mM KCl, 0.5 mM $MgCl_2$, 10% glycerol, 0.02% NP-40, 1% glucose, 0.1 mg/mL acetylated BSA, 2 mM Trolox, 0.03 mM β–mercaptoethanol, 2 U/µL catalase, and 0.08 U/µL glucose oxidase. SNF2h and ADP-$BeF_x$ were added simultaneously to a final concentration of 2 µM and 0.5 mM respectively using an automated syringe pump. Nucleosomes were then imaged after a 10 min incubation to match the EM preparation conditions. 7 min movies were collected at a sufficiently high laser power that most nucleosomes photobleached before the end of the movie, enabling the exclusion of nucleosomes that did not exhibit single-step photobleaching in both channels. The reported FRET value for each nucleosome is the average over the portion of the movie prior to the first photobleaching event.

## ATPase assays

DNA stimulated ATPase assays were performed using an NADH coupled assay (*Lindsley, 2001*). Reactions were performed with 800 nM SNF2h, saturating concentrations of 207 bp DNA (208 nM) and ATP-$MgCl_2$ (4 mM) with 10 U/µL lactate dehydrogenase, 10 U/µL Pyruvate kinase, 180 µM NADH, 2 mM phosphoenol pyruvate, 12.5 mM HEPES pH 7.5, 70 mM KCl, 3 mM free $MgCl_2$, and ~1.5% glycerol at 25˚C. Reactions were incubated in a 384 well plate at 25˚C prior to addition of enzyme to initiate the reaction. Absorbance was monitored at 340 nm in a SpectraMax M5e plate reader and the resulting data was background subtracted using absorbance at 420 nm. The linear phase of each reaction was then fit using linear regression using Prism to obtain hydrolysis rates. For nucleosome stimulated ATP hydrolysis, rates were measured using radioactivity as in *Gamarra et al. (2018)* under saturating concentrations of nucleosomes without flanking DNA and subsaturating concentrations of (20 µM) ATP-$MgCl_2$. Reactions were performed in 12.5 mM HEPES pH 7.5, 70 mM KCl, 3 mM free $MgCl_2$, 0.02% NP-40, and ~1.5% glycerol at 25˚C, initiated by addition of enzyme, and time points quenched with an equal volume 50 mM Tris pH 7.5, 3% SDS, and 100 mM EDTA. Time points were resolved on a PEI-cellulose TLC plate (Select Scientific) using a 0.5M LiCl/1M Formic acid mobile phase, plates were dried and then exposed on a phosphorscreen overnight. The screen was imaged using a Typhoon variable mode imager. Fraction of ATP hydrolyzed was quantified using ImageJ and initial rates were determined by fitting a line through the first 10% of inorganic phosphate generated using Prism.

## Acknowledgements

We thank J Tretyakova for tirelessly purifying histone proteins and help with generating DNA substrates, D Asarnow for providing the script for variance map calculation and signal subtraction and G Bowman and RF Levendosky for sharing results from unpublished data. This work is supported by grants from the NIH (GM073767 and GM108455 to GJN and R01GM082893 and 1S10OD020054 to YC) and the UCSF Program for Breakthrough Biomedical Research (New Technology Award) to YC. NG is supported by a pre-doctoral fellowship from the NSF. SLJ was supported by a Leukemia and Lymphoma Society Career Development Fellow award. YC is an Investigator with the Howard Hughes Medical Institute.

## Additional information

### Competing interests

Geeta J Narlikar: Reviewing editor, *eLife*. John D Leonard: Is affiliated with 3T Biosciences and has no other competing interests to declare. The other authors declare that no competing interests exist.

### Funding

| Funder | Grant reference number | Author |
| --- | --- | --- |
| National Institutes of Health | GM073767 | Geeta J Narlikar |
| National Science Foundation | | Nathan Gamarra |
| Howard Hughes Medical Institute | | Yifan Cheng |
| University of California, San Francisco | Program for Breakthrough Biomedical Research | Yifan Cheng |
| Leukemia and Lymphoma Society | | Stephanie L Johnson |
| National Institutes of Health | GM108455 | Geeta J Narlikar |
| National Institutes of Health | R01GM082893 | Yifan Cheng |
| National Institutes of Health | 1S10OD020054 | Yifan Cheng |

The funders had no role in study design, data collection and interpretation, or the decision to submit the work for publication.

### Author contributions

Jean Paul Armache, Conceptualization, Resources, Data curation, Software, Formal analysis, Validation, Investigation, Visualization, Methodology, Writing—original draft, Writing—review and editing; Nathan Gamarra, Conceptualization, Resources, Data curation, Formal analysis, Validation, Investigation, Visualization, Methodology, Writing—original draft, Writing—review and editing; Stephanie L Johnson, Conceptualization, Resources, Data curation, Software, Formal analysis, Validation, Investigation, Visualization, Methodology, Writing—review and editing; John D Leonard, Conceptualization, Resources, Investigation, Methodology, Writing—review and editing; Shenping Wu, Investigation, Visualization, Methodology, Writing—review and editing; Geeta J Narlikar, Yifan Cheng, Conceptualization, Supervision, Funding acquisition, Methodology, Writing—original draft, Project administration, Writing—review and editing

### Author ORCIDs

Jean Paul Armache (iD) https://orcid.org/0000-0001-9195-2282
Nathan Gamarra (iD) https://orcid.org/0000-0002-2430-8662
Geeta J Narlikar (iD) https://orcid.org/0000-0002-1920-0147
Yifan Cheng (iD) https://orcid.org/0000-0001-9535-0369

**Decision letter and Author response**
Decision letter https://doi.org/10.7554/eLife.46057.045
Author response https://doi.org/10.7554/eLife.46057.046

# Additional files

## Supplementary files

• Supplementary file 1. Key Resources Table.
DOI: https://doi.org/10.7554/eLife.46057.029

• Transparent reporting form
DOI: https://doi.org/10.7554/eLife.46057.030

## Data availability

The cryo-EM density maps have been deposited in the Electron Microscopy Data Bank (EMDB) under accession numbers EMD-9353 (nucleosome with doubly bound SNF2h), EMD-9351 (nucleosome with singly bound SNF2h at SHL+2, 8.4Å), EMD-9352 (SNF2h-nucleosome, 3.4Å), EMD-9354 (SNF2h-nucleosome SHL-2, 3.9Å), and EMD-9355 (SNF2h-nucleosome SHL+2, 6.9Å) (Figure 2-supplement 4). Corresponding unsharpened map and both half maps were deposited as maps associated with the primary depositions. Particle image stacks of nucleosome with singly bound SNF2h after motion correction have been deposited in the Electron Microscopy Public Image Archive (http://www.ebi.ac.uk/pdbe/emdb/empiar/) under accession number EMPIAR-341. Atomic coordinate of nucleosome with SNF2h bound near entry side has been deposited in the Protein Data Bank (PDB) under the accession number 6NE3.

The following datasets were generated:

| Author(s) | Year | Dataset title | Dataset URL | Database and Identifier |
|---|---|---|---|---|
| Jean Paul Armache, Nathan Gamarra, Stephanie L Johnson, John D Leonard, Shenping Wu, Geeta J Narlikar, Yifan Cheng | 2019 | Nucleosome with singly bound SNF2h at SHL+2, 8.4Å | http://www.ebi.ac.uk/pdbe/entry/emdb/EMD-9351 | EMDataBank, EMD-9351 |
| Jean Paul Armache, Nathan Gamarra, Stephanie L Johnson, John D Leonard, Shenping Wu, Geeta J Narlikar, Yifan Cheng | 2019 | SNF2h-nucleosome, 3.4Å | http://www.ebi.ac.uk/pdbe/entry/emdb/EMD-9352 | EMDataBank, EMD-9352 |
| Jean Paul Armache, Nathan Gamarra, Stephanie L Johnson, John D Leonard, Shenping Wu, Geeta J Narlikar, Yifan Cheng | 2019 | nucleosome with doubly bound SNF2h | http://www.ebi.ac.uk/pdbe/entry/emdb/EMD-9353 | EMDataBank, EMD-9353 |
| Jean Paul Armache, Nathan Gamarra, Stephanie L Johnson, John D Leonard, Shenping Wu, Geeta J Narlikar, Yifan Cheng | 2019 | SNF2h-nucleosome SHL-2, 3.9Å | http://www.ebi.ac.uk/pdbe/entry/emdb/EMD-9354 | EMDataBank, EMD-9354 |
| Jean Paul Armache, Nathan Gamarra, Stephanie L Johnson, John D Leonard, Shenping Wu, Geeta J Narlikar, Yifan Cheng | 2019 | SNF2h-nucleosome SHL+2, 6.9Å | http://www.ebi.ac.uk/pdbe/entry/emdb/EMD-9355 | EMDataBank, EMD-9355 |

| Jean Paul Armache, Nathan Gamarra, Stephanie L Johnson, John D Leonard, Shenping Wu, Geeta J Narlikar, Yifan Cheng | 2019 | Nucleosome with singly bound SNF2h after motion correction | http://www.ebi.ac.uk/pdbe/emdb/empiar/entry/341 | Electron Microscopy Public Image Archive, EMPIAR-341 |
| Jean Paul Armache, Nathan Gamarra, Stephanie L Johnson, John D Leonard, Shenping Wu, Geeta J Narlikar, Yifan Cheng | 2019 | Cryo-EM structure of singly-bound SNF2h-nucleosome complex at 3.4 A | http://www.rcsb.org/structure/6NE3 | Protein Data Bank, 6NE3 |

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
