## [Decision Letter]

Thank you for submitting your article "Cryo-EM structures of remodeler-nucleosome intermediates suggest allosteric control through the nucleosome" for consideration by *eLife*. Your article has been reviewed by three peer reviewers, including Sjors HW Scheres as the Reviewing Editor and Reviewer #1, and the evaluation has been overseen by Cynthia Wolberger as the Senior Editor.

The reviewers have discussed the reviews with one another and the Reviewing Editor has drafted this decision to help you prepare a revised submission. In light of the concerns expressed regarding competition, we note that *eLife* provides 'scoop-protection', which means that if another paper comes out while the revision is being done, this does not affect the decision to publish in *eLife*. In light of the competitive situation, you are welcome to communicate with the Senior Editor with any questions regarding the points raised below in advance of preparing a revised manuscript.

Summary:

This manuscript presents data indicating structural changes to the histone components of the nucleosome (in addition to the DNA) upon binding of the remodelling enzyme SNF2h in the presence of ADP-BeF_x_. This potentially represents key information to support previous data generated by the Narlikar lab which showed that the action of SNF2h was sensitive to cross-links introduced at specific locations within the histone octamer. The three reviewers agreed this manuscript contains sufficiently interesting data to warrant publication in *eLife*. However, there was also a consensus that the 'low-salt' data set is of insufficient quality to merit the conclusions drawn on it (see point 1). Therefore, the authors are encouraged to include a new structure from the sample in the low-salt condition using a direct-electron detector in a revised version. If this is not possible at present, the 2 bp translocation detected in the 3.9Å structure is still a novel observation, and it should be possible to produce a very important manuscript focusing on this aspect alone.

Essential revisions:

1) The observation that the nucleosome becomes partially disordered when two SNF2h molecules bind in the low-salt condition plays an important role in the Discussion section. This observation, in the 8.4Å dimer map, is not supported by the data: at these resolutions disorder is very hard to distinguish from noise present in the reconstruction. Detailed analysis of the uploaded maps confirms this. Therefore, the authors are encouraged to include a new structure, calculated from a data set recorded on a direct-electron detector, in a revised version of this manuscript. Otherwise, the model in Figure 5 should be substantially modified.

2) The sub-division of the K2 data set is not clear: there is a separate path of image processing yielding a 3.4Å resolution map of the SNF2h bound at the -2 position, whereas a split of the same data set into +2 and -2 bound positions yield a 3.9Å and a 6.9Å map. Could it be that the 3.4Å map contains a mixture of the two states, and despite a higher resolution, the 3.4Å map is actually representing a (suboptimal) mixture? All particles contributing to the 6.9Å map should be removed from the 3.4Å map.

Other points:

Several different datasets are used and different routes used to process these (Figure 2—figure supplement 2). These illustrate the difficulty in distinguishing between particles in which SNF2h is bound to one side of the nucleosome or the other. Subtraction of the density for SNF2h, alignment to linker DNA and back projection was required to distinguish between SNF2H is bound at superhelical location -2 and +2. (Figure 2—figure supplement 2B). In the text it is stated that a similar process was used to orient doubly bound nucleosomes from the lower resolution dataset. However, this is not indicated in Figure 2—figure supplement 2A). This should be clarified. Was this applied for the singly bound species at 70mM KCl? Why is there no SNF2H bound at SHL -2 at 70 mM KCl?

The density for SNF2h at SHL+2 is lower resolution than that at SHL-2 this is interpreted as increased conformational flexibility. Is it possible that this instead arises because some of the particles included are singly bound on the SHL-2 side?

Presumably the 2 bp translocation is also observed in the 3.4Å structure?

In the Materials and methods, it is specified that the low salt sample contained 0.5mM MgCl_2_ in addition to 70mM KCl. But in the "higher salt" sample [140mM KCl], there was no MgCl_2_ added. In terms of chromatin compaction, 0.5mM of divalent cations would correspond to approximately 50mM of monovalent salt. Could the difference between the two datasets be related to Mg concentration rather than the amount of KCL?

Since particles may dissociate during grid preparation, a gel shift EMSA analysis using gentle fixation would be more informative than 2D/3D classification to analyse stoichiometry.

The single molecule FRET experiments provide a useful alternate means of measurement to support the SNF2h dependent extension of linker DNA. Is it known which side of the nucleosome SNF2h is acting from in these experiments? This important to know whether it substantiates the DNA extension observed in the +2 or -2 structure.

Could the ensemble FRET shown in Figure 1—figure supplement 3 be affected by changes in the DNA trajectory rather than DNA translocation? The explanation for the rapid reduction in FRET on adding SNF2h and nucleotide is not clear. The single molecule experiments provide much stronger evidence for c2bp translocation, however, in this case the measurement at 70 mM KCl was not made. It would be good to add this.

"During the course of this study, we collected two cryo-EM datasets using two different salt conditions for optimization of cryo-EM grid preparation." In terms of optimization of grid preparation, where there more than two conditions screened?

"Particles contributing to this reconstruction were aligned unambiguously using flanking DNA as a fiducial marker to break the pseudo symmetry (Figure2A, Figure 2—figure supplement 2A).” But given the low resolution/low SNR of this data set, this may not be true. There are in fact many views where the extra DNA is not visible.

"The density of the SNF2h bound at SHL+2 is weaker than that of SNF2h bound at SHL-2 (Figure2A, Figure 2—figure supplement 2A). This difference suggests that the SNF2h bound to the nucleosome at SHL+2 is conformationally more flexible." How can one exclude that instead of flexibility, the worse density of the second SNF2h is caused by incomplete dimer formation, i.e. partial occupancy of the SNF2h at the +2 position? (Again better resolution might help to distinguish between the two, see major point 1.)

"DNA translocation was not detected in the EM reconstructions carried out under the 70 mM KCl conditions" yet later it is stated "Unlike the structures at 70 mM KCl, the structures obtained at 140 mM KCl show a higher proportion of singly bound SNF2h at SHL-2 compared to SHL+2. Yet all the nucleosomes show the 2 bp DNA translocation in the same direction." I don't understand this. What does "all nucleosomes" mean? In both the high and low salt conditions?

Figure 1—figure supplement 6: it is not clear at which threshold this map is shown. In fact, no orange density is visible at all. It would be useful to see the variance thresholded at for example 2 or 3 sigma, or to show a few slices with grey-scale images through the variance map.

Figure 1D & Figure 3—figure supplement 2: what is shown in purple? SNF2h? Perhaps it's helpful to add more labels for residues/chains.

The field view images are too small and at too low resolution to clearly see, but it looks like there are a lot of larger complexes (containing more than one "singly bound" or "doubly bound "nucleosome). That is also seen in the negative stain images, but not discussed in the paper.

The uploaded maps for the low-salt reconstructions both show evidence that multiple spherical masks (or the 'Volume Eraser' tool in UCSF Chimera?) have been used to remove density. What was in these positions before? Unmasked maps should be uploaded to the EMDB.

Electron micrographs are too small in the images to see the particles clearly

Referring to datasets as "early" and "recent" is slightly confusing, especially in the beginning of the Results section.

---

## [Author Response]

Essential revisions:1) The observation that the nucleosome becomes partially disordered when two SNF2h molecules bind in the low-salt condition plays an important role in the Discussion section. This observation, in the 8.4Å dimer map, is not supported by the data: at these resolutions disorder is very hard to distinguish from noise present in the reconstruction. Detailed analysis of the uploaded maps confirms this. Therefore, the authors are encouraged to include a new structure, calculated from a data set recorded on a direct-electron detector, in a revised version of this manuscript. Otherwise, the model in Figure 5 should be substantially modified.

We appreciate and understand the fact that our reconstruction of nucleosome bound with two SNF2h is not ideal. The major challenge of obtaining a cryo-EM structure of nucleosome-SNF2h complex was in sample preparation. Without cross-linking, cryo-EM grid preparation is very unpredictable and we have not been able to consistently obtain suitable cryo-EM grids of nucleosomes bound with two SNF2h. From many trials in the past several years, we are convinced that the problem most likely arises from particles dissociating at the air-water interface specifically during cryo-EM grid preparation, as we can repeatedly and reproducibly observe nucleosomes with two SNF2h bound by negative stain EM, but not by cryo-EM. We are still trying, but unfortunately, we cannot produce a new structure within the time frame of this revision.

Regarding our interpretation, we respectfully disagree with reviewer’s comment. Indeed, 8.4Å is a relative low resolution and by itself the weaker density of one helix cannot be interpreted as implying more disorder. However, we compare the weaker density of one helix with the stronger density of its “symmetry” related helix within the same SNF2h-nucleosome complex. Because the “symmetry” related helix serves as an internal control and comparison, we propose the weaker density of the helix can be interpreted as implying disorder. Furthermore, our interpretation is also supported by biochemical and NMR results from our previous published studies (Sinha et al., 2017).

Nevertheless, we do recognize that the relative low-resolution of the reconstruction casts some uncertainty on our interpretation. Following the suggestions of the reviewers, we have modified the model in Figure 5, and we now describe the disordered state as speculative and as providing a starting point for understanding the structural basis of SNF2h mediated octamer fluctuations. Analogous changes have also been made in the main text (e.g. in the second paragraph of the Discussion).

2) The sub-division of the K2 data set is not clear: there is a separate path of image processing yielding a 3.4Å resolution map of the SNF2h bound at the -2 position, whereas a split of the same data set into +2 and -2 bound positions yield a 3.9Å and a 6.9Å map. Could it be that the 3.4Å map contains a mixture of the two states, and despite a higher resolution, the 3.4Å map is actually representing a (suboptimal) mixture? All particles contributing to the 6.9Å map should be removed from the 3.4Å map.

We appreciate this comment from the reviewers.

We processed this dataset twice, independently. The first processing yielded a clear division into two subsets: 3.9Å and 6.9Å. After releasing an improved version of MotionCor2, we reprocessed the entire dataset in every step, from motion correction, particle picking, classification and refinement. This re-processing produced a map with improved resolution of 3.4Å. We did not attempt again to re-classify the second particle stack (re-picked from the same raw micrographs). Indeed, the 3.4Å reconstruction contains a mixture of two states, nucleosome with SNF2h bound at +2 and -2 positions.

In the revision, we performed a further 3D classification of the 3.4Å dataset into two classes (Figure 2—figure supplement 2): a clearly defined 3.6Å SHL-2 reconstruction from ~2/3 of the particles in the 3.4Å reconstruction and another 4.5Å reconstruction from remaining ~1/3 of the particles. The latter 4.5Å reconstruction likely still contains a mix of SHL+2 and SHL-2 that we were unable to separate further. The 3.6Å map very closely resembles the 3.4Å reconstruction, with no change in interpretation. In the revised manuscript, we have updated the figures, and revised the Materials and methods section to make this clear (subsection “Validation of 2 base-pair translocated nucleosome”).

Other points:

Several different datasets are used and different routes used to process these (Figure 2—figure supplement 2). These illustrate the difficulty in distinguishing between particles in which SNF2h is bound to one side of the nucleosome or the other. Subtraction of the density for SNF2h, alignment to linker DNA and back projection was required to distinguish between SNF2H is bound at superhelical location -2 and +2. (Figure 2—figure supplement 2B). In the text it is stated that a similar process was used to orient doubly bound nucleosomes from the lower resolution dataset. However, this is not indicated in Figure 2—figure supplement 2A). This should be clarified. Was this applied for the singly bound species at 70mM KCl? Why is there no SNF2H bound at SHL -2 at 70 mM KCl?

We now clarified this procedure in Figure 2—figure supplement 2. For singly bound particles at 70mM KCl, the same procedure is applied and yielded only one reconstruction with SNF2h bound at SHL+2 position. Although we cannot completely rule out the possibility of having any particles with SNF2h bound at SHL-2 position included in the final reconstruction, such particles must be very few if any. We speculate that the absence of SHL-2 particles may be due to a weaker affinity of this protomer for the nucleosome in the 70mM KCl condition. The weaker affinity of SHL-2 compared to SHL+2 SNF2h could be the result of multiple aspects of the nucleosome substrate. This includes the absence of flanking DNA on the exit side of the nucleosome, which could be bound by the HSS of the SHL-2 protomer. Additionally, asymmetric dynamics in the H2A/H2B acidic patch, a surface that contributes to SNF2h binding at 70mM KCl (Gamarra et al. 2018, PMID: 29664398), could specifically destabilize binding by SHL-2 SNF2h. Further, recent studies by the Bowman lab (Levendosky et al. 2019, PMID: 31094676), suggest that the SHL-2 protomer uses the acidic patch on face A of the histone octamer to activate nucleosome sliding. This is the same surface that appears to be dynamic in our cryo-EM reconstructions. Consistent with this possibility, we observe a higher proportion of SHL-2 bound species in the 140mM KCl condition where we do not detect histone dynamics. The combination of these features may specifically weaken the interaction of the SHL-2 protomer with the nucleosome at 70mM KCl, resulting in its preferential disassociation from the nucleosome.

The density for SNF2h at SHL+2 is lower resolution than that at SHL-2 this is interpreted as increased conformational flexibility. Is it possible that this instead arises because some of the particles included are singly bound on the SHL-2 side?

Separation of singly and doubly bound particles is actually quite effective and straightforward. Therefore, we do not think the weaker density of one SNF2h is caused by including singly bound particles. While we are confident about this, we do agree that it is impossible to completely rule out such possibility. We revised the text as “This difference likely suggests that […]”.

Presumably the 2 bp translocation is also observed in the 3.4Å structure?

Yes, it is clearly there. The 3.9Å and 3.4Å, as well as 6.9Å densities, all contain the 2bp translocation-indicating density.

In the Materials and methods, it is specified that the low salt sample contained 0.5mM MgCl_2_ in addition to 70mM KCl. But in the "higher salt" sample [140mM KCl], there was no MgCl_2_ added. In terms of chromatin compaction, 0.5mM of divalent cations would correspond to approximately 50mM of monovalent salt. Could the difference between the two datasets be related to Mg concentration rather than the amount of KCL?

This was a typo on our part. We apologize for this oversight. Both the low and higher salt conditions contained the same amount of MgCl_2_. We have corrected this omission in the revised manuscript.

Since particles may dissociate during grid preparation, a gel shift EMSA analysis using gentle fixation would be more informative than 2D/3D classification to analyse stoichiometry.

The reviewer raises a good point. Indeed we have previously carried out detailed studies of stoichiometry as described in (Racki et al. 2009, PMID: 20033039). In this paper we (i) measured the hill coefficient for activity and binding, which is close to 2, (ii) used EPR with spin labels on the histone H4 tail and found that both tails were bound implying two SNF2h subunits and (iii) visually assessed stoichiometry by negative stain EM, which clearly showed that a majority of SNF2h-nucleosome complexes contained two SNF2h molecules bound to a single nucleosome. In this current study we have reproduced our negative stain EM results from 2009 (Figure 2—figure supplement 3), giving confidence that the stoichiometry in solution is 2 SNF2h protomers bound to a single nucleosome.

However, as the reviewer suggests, some particles are clearly dissociating during cryo-EM grid preparation under the higher salt conditions as we do not observe doubly bound SNF2h particles. While cross-linking provides a way to stabilize the complex, our goal was to trap conformational changes in SNF2h and the nucleosome and we were worried that cross-linking would alter such states.

The single molecule FRET experiments provide a useful alternate means of measurement to support the SNF2h dependent extension of linker DNA. Is it known which side of the nucleosome SNF2h is acting from in these experiments? This important to know whether it substantiates the DNA extension observed in the +2 or -2 structure.

Substantial previous work using gaps and nicks in DNA (Schwanbeck et al. 2004 PMID: 15262970, Zofall et al. 2006 PMID: 16518397) has indicated that ATP-dependent movement of DNA from the exit site is catalyzed by the SNF2h motor bound at SHL+2. We therefore hypothesize that the movement observed in our single molecule experiments is also occurring from SNF2h with its ATPase bound at SHL+2.

Could the ensemble FRET shown in Figure 1—figure supplement 3 be affected by changes in the DNA trajectory rather than DNA translocation? The explanation for the rapid reduction in FRET on adding SNF2h and nucleotide is not clear. The single molecule experiments provide much stronger evidence for c2bp translocation, however, in this case the measurement at 70 mM KCl was not made. It would be good to add this.

We agree with the reviewer that the ensemble FRET could also be affected by changes in DNA trajectory. In terms of the rapid reduction in FRET upon adding SNF2h, we have previous shown that this is due to an environmental effect of SNF2h on the dye fluorescence (Racki et al. 2009 PMID: 20033039, Leonard and Narlikar 2014 PMID: 25684208). In these previous studies we showed nucleosomes that only contained a Cy3 label on the DNA exit site displayed an increase in Cy3 fluorescence upon SNF2h binding that was ATP-independent and could be used to obtain Kd values. We therefore interpret the rapid decrease in FRET in Figure 1 supplement 3 as arising from this environmental effect, which enhances Cy3 fluorescence. The decrease in FRET at later times we interpret as arising from a change in the location of the DNA with respect to the histone core. We have now clarified this reasoning in the revised manuscript and also now mention the possibility that DNA trajectory can contribute to the change in FRET (Section “A SNF2h-nucleosome complex with a translocated nucleosome”, Figure 1—figure supplement 3C-D legend).

In terms of carrying out smFRET experiments at 70 mM KCl, we agree these would be quite informative. However the technical challenges associated with these experiments makes them not feasible within the time-frame of the revision.

"During the course of this study, we collected two cryo-EM datasets using two different salt conditions for optimization of cryo-EM grid preparation." In terms of optimization of grid preparation, where there more than two conditions screened?

We have repeatedly obtained negative stain EM images showing the majority of particles are nucleosomes with two SNF2h protomers bound. The biochemical conditions are well-optimized for creating doubly bound SNF2h-nucleosome complexes, but we were able to capture such conditions in cryo-EM only once. We have tried all available approaches in cryo-EM grid preparation. As mentioned above, chemical cross-linking can, in principle, preserve the doubly bound complexes but we avoided using this approach due to concerns that it may lock the structure in a conformation that may not be biologically relevant.

"Particles contributing to this reconstruction were aligned unambiguously using flanking DNA as a fiducial marker to break the pseudo symmetry (Figure2A, Figure 2—figure supplement 2A).” But given the low resolution/low SNR of this data set, this may not be true. There are in fact many views where the extra DNA is not visible.

We appreciate the comment, and removed the word “unambiguously”, recognizing that it is impossible to completely rule out any alignment error.

"The density of the SNF2h bound at SHL+2 is weaker than that of SNF2h bound at SHL-2 (Figure2A, Figure 2—figure supplement 2A). This difference suggests that the SNF2h bound to the nucleosome at SHL+2 is conformationally more flexible." How can one exclude that instead of flexibility, the worse density of the second SNF2h is caused by incomplete dimer formation, i.e. partial occupancy of the SNF2h at the +2 position? (Again better resolution might help to distinguish between the two, see major point 1.)

As responded above, separating particles of singly or doubly bound SNF2h is rather efficient. Nevertheless, we do appreciate this comment and recognize the separation is not completely certain. In the revised manuscript, we add the word “likely” to reflect the fact that there is some, although small, uncertainty in our statement.

"DNA translocation was not detected in the EM reconstructions carried out under the 70 mM KCl conditions" yet later it is stated "Unlike the structures at 70 mM KCl, the structures obtained at 140 mM KCl show a higher proportion of singly bound SNF2h at SHL-2 compared to SHL+2. Yet all the nucleosomes show the 2 bp DNA translocation in the same direction." I don't understand this. What does "all nucleosomes" mean? In both the high and low salt conditions?

We modified the sentence to clarify this: “Yet all the nucleosomes in the higher salt conditions show the 2 bp DNA translocation in the same direction."

Figure 1—figure supplement 6: it is not clear at which threshold this map is shown. In fact, no orange density is visible at all. It would be useful to see the variance thresholded at for example 2 or 3 sigma, or to show a few slices with grey-scale images through the variance map.

In the full quality image, the variance map is defined as orange and barely visible in the image as the bootstrapped maps are all nearly identical. However, per reviewer’s request, we modified this figure to contain variance map at sigma levels 2, 3, 4 and 5.

Figure 1D & Figure 3—figure supplement 2: what is shown in purple? SNF2h? Perhaps it's helpful to add more labels for residues/chains.

We have revised this figure according to the suggestions.

The field view images are too small and at too low resolution to clearly see, but it looks like there are a lot of larger complexes (containing more than one "singly bound" or "doubly bound "nucleosome). That is also seen in the negative stain images, but not discussed in the paper.

We now address this issue explicitly in the revised manuscript. Specifically we say: “Only monodispersed particles are picked. Particle aggregations, which are occasionally seen, are excluded.”

The uploaded maps for the low-salt reconstructions both show evidence that multiple spherical masks (or the 'Volume Eraser' tool in UCSF Chimera?) have been used to remove density. What was in these positions before? Unmasked maps should be uploaded to the EMDB.

We apologize that we uploaded a wrong map. The correct map is now uploaded for reviewers to evaluate. We have now double checked and confirmed that all figures are prepared with the correct and unmodified map.

Electron micrographs are too small in the images to see the particles clearly

We have increased the size as suggested.

Referring to datasets as "early" and "recent" is slightly confusing, especially in the beginning of the Results section.

We have changed the relevant text to remove the words “early” and “recent” and now only describe the two data sets based on how they were collected.